# Evolution of acoustic communication in blind cavefish

Carole Hyacinthe[1], Joël Attia[2] & Sylvie Rétaux [1]

Acoustic communication allows the exchange of information within specific contexts and during specific behaviors. The blind, cave-adapted and the sighted, river-dwelling morphs of the species *Astyanax mexicanus* have evolved in markedly different environments. During their evolution in darkness, cavefish underwent a series of morphological, physiological and behavioral changes, allowing the study of adaptation to drastic environmental change. Here we discover that *Astyanax* is a sonic species, in the laboratory and in the wild, with sound production depending on the social contexts and the type of morph. We characterize one sound, the "Sharp Click", as a visually-triggered sound produced by dominant surface fish during agonistic behaviors and as a chemosensory-, food odor-triggered sound produced by cavefish during foraging. Sharp Clicks also elicit different reactions in the two morphs in play-back experiments. Our results demonstrate that acoustic communication does exist and has evolved in cavefish, accompanying the evolution of its behaviors.

[1] Paris-Saclay Institute of Neuroscience, CNRS UMR9197, Université Paris-Saclay, Gif-sur-Yvette, France. [2] Equipe Neuro-Ethologie Sensorielle ENES/Neuro-PSI CNRS UMR9197, Université de Lyon/Saint-Etienne, Saint-Etienne, France. Correspondence and requests for materials should be addressed to S.R. (email: retaux@inaf.cnrs.-gif.fr)

Acoustic signals are widely used among animals for multiple communication and behavioral purposes, including in the aquatic environment[1,2]. Bony fishes have evolved diverse sound generating mechanisms that are well-studied[3–5] and use them for intraspecific communication, often for agonistic and mating behaviors[6–9]. However, how acoustic communication can evolve via adaptation within species with respect to needs in their specific environments is unknown. To address this question we have used the two morphs of the species Astyanax mexicanus. The river-dwelling sighted form and the cave-adapted blind form, which diverged about 20,000 years ago[10], have since then experienced markedly different habitats and environmental pressures[11–13]. During their adaptation to perpetual darkness, cavefish underwent a series of morphological, physiological, and behavioral changes[14–19], including major modifications in their chemosensory and mechanosensory systems which help them to navigate, find food and find mates in the dark[20–24]. Yet, hearing abilities are similar in the two morphs[25] and to our knowledge, nothing is known about acoustic communication in A. mexicanus.

Here, we question whether acoustic communication has evolved differently in the two morphs of A. mexicanus and accompanied behavioral shifts in the absence of visual communication in cavefish. We show that A. mexicanus is a highly sonic species, in the laboratory and in the wild, with a repertoire of at least 6 sounds. When studied in controlled laboratory conditions, sound production quantitatively and qualitatively depends on the social contexts (solo, duo, or group) and the type of morph. We then further characterize one sound, the "Sharp Click", as a visually-triggered sound produced by dominant surface fish during agonistic behavior and as a chemosensory-triggered sound produced by cavefish during foraging behavior, and which also elicits different reactions in the two morphs in play-back experiments.

## Results

### Astyanax mexicanus is a sonic species, in the lab and in the wild.
First we determined whether the species Astyanax mexicanus produces sounds. Six different types of sounds could be identified from 60 h of acoustic recordings of adult A. mexicanus surface fish (SF) or Pachón cavefish (CF) in the laboratory, including three simple sounds and three complex sounds (Fig. 1a–f and Supplementary Audio 1–6; Supplementary Fig. 1a and Methods). Clocs, Clicks, and Sharp Clicks corresponded to single pulses of short duration (lasting < 20 ms, separated by > 1 s interval from the next pulse; Table 1). Serial Clocs, Serial Clicks, and Rumblings corresponded to repetitions of single pulses and to longer sounds lasting up to 1 s, respectively. Each sound type had a specific structure and spectral range and showed maximum energy in specific frequencies (Table 1 and Fig. 1a–f), which are typically in the range of A. mexicanus hearing capabilities tested with a behavioral assay between 50 and 7500 Hz, with a maximal sensitivity around 1000 Hz[25].

To validate the identification and classification of the sounds produced by A. mexicanus, principal component analyses (PCA) were performed after extraction of acoustic parameters for the simple sounds (total 516 sounds analyzed) and after pulse rate analysis for the complex sounds (total 186 sounds analyzed). First, the three simple sound types, i.e., Clocs, Clicks, and Sharp Clicks, grouped into clusters and were thus confirmed as distinct categories of sounds, for both morphs (Fig. 1g and Supplementary Data 1). Moreover, a pDFA (permutated Discriminant Function Analysis)[26] on the same dataset generated a confusion matrix with good scores of correctly reclassified sounds (Fig. 1h). Second, Serial Clicks and Serial Clocs also belonged to separate clusters on

the grounds of temporal parameters, and also for both morphs (Fig. 1i, j and Supplementary Data 2). These results support the hypothesis that the different simple sounds and complex sounds identified could carry different information and could be used differently according to context and behavior. Overall these data indicated that A. mexicanus is a sonic species, and that the surface and cave morphs share a repertoire of 6 simple or complex sounds. Of note, some inter-morph differences existed in the detailed acoustic parameters of the three simple sounds, especially for Single Clocs (Table 1), as well as in the fine pulse rate parameters for the Serial Clocs (Supplementary Fig. 2).

To ascertain the biological and potential adaptive relevance of A. mexicanus sound production, we next sampled the two morphs in their natural environment. Among the 30 caves with cavefish populations in Mexico[11,12], we visited 6 locations: Molino, Pachón, Los Sabinos, Tinaja, Chica, and Subterráneo (Fig. 2a). These caves are representative of the 3 proposed independent colonization events by ancestral surface fish populations in the subterranean milieu of the Sierra de El Abra, Sierra de Guatemala, and Sierra de Colmena, respectively[27]. In these 6 caves, and in a well in which surface morphs are found, sounds alike those identified in the lab were recorded (Fig. 2a, for Serial Clicks-like and Supplementary Audio 7–13 for Serial Clicks in the 6 caves; Fig. 2b and Supplementary Audio 8, 14–19 for the 6 sounds in the Pachón cave). Furthermore, a PCA performed on acoustic parameters extracted from single Clicks and Single Clocs recorded in the lab (total of 336 sounds analyzed) and in the wild in the Pachón cave (total 89 sounds analyzed) demonstrated that (1) in the wild also, Single Clicks and Single Clocs were easily discernable and corresponded to distinct sounds, and (2) sounds produced in the wild were alike those produced in laboratory conditions. In sum, the repertoire of six sounds was shared in independently-evolved Astyanax cave and surface populations, despite diverse local environmental conditions (e.g., degree of isolation, nutrient availability, seasonal changes, or presence of predators). We thus pursued additional analyses of sounds in controlled laboratory conditions.

The major differences between the two A. mexicanus morphs, which are relevant to inter-individual communication, are (1) the apparent lack of social structure and schooling behavior associated to an absence of hierarchical aggressiveness in cavefish[16,19,28–31], and (2) the absence of visual modality in blind cavefish, causing them to be more reliant on other sensory systems. We then addressed the changes in acoustic communication and associated behaviors in cavefish, with regards to these two major differences.

### Sound production and social interactions.
We predicted that, if sounds produced by A. mexicanus are used for acoustic communication, then production should vary according to the social context. We compared sound production in individual fish (solo), pairs of fish (duo), and groups of 6 fish (group; SF or Pachón CF) during the exploration of a new environment and new conspecific (s), or after habituation. Ethograms, i.e., graphs depicting the production of sounds or behaviors along time, were generated (Fig. 3a, b). As it was impossible to know which individual emitted sounds when more than one fish was present in the tank, sound production was normalized per fish and per time in the following analyses. In the solo condition, Pachón emitted more sounds than SF for almost all sound types, resulting in a 3 fold higher total number of sounds produced in 30 min (Fig. 3c, e–j). In the duo context, Pachón CF produced significantly less sounds and SF increased their sound production as compared to solo, hence the two morphs emitted a similar number of total sounds. Finally, in groups, SF and CF emitted a moderate quantity of

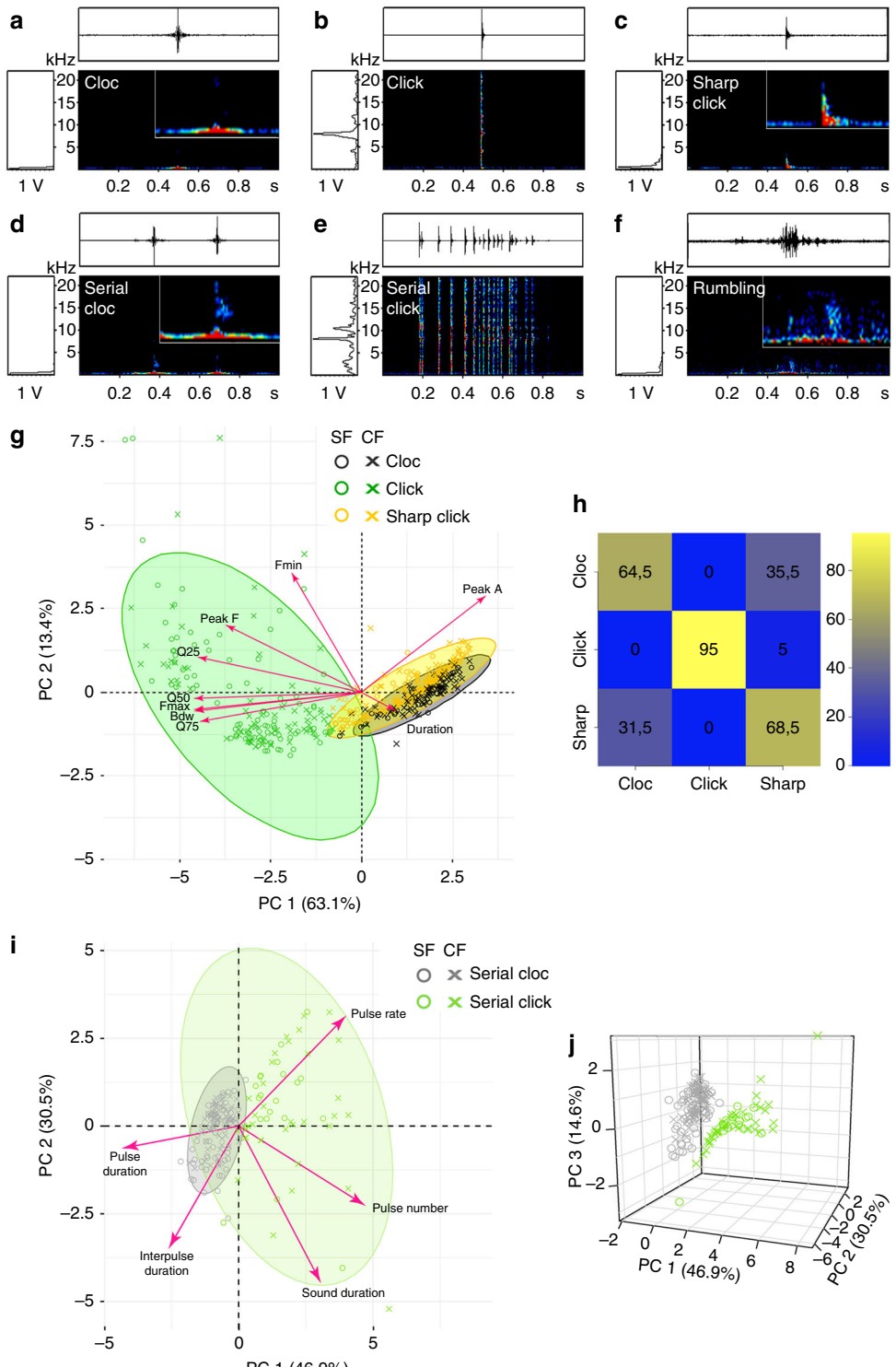

**Fig. 1** *A. mexicanus* is a sonic species in the lab. **a–f** Representative sonograms of the 6 types of sounds identified. Magnifications (×3) on a 0.2 s window are provided as insets, to appreciate details of the sonogram for all sounds that do not spread to frequencies higher than 5 kHz (i.e., only Clicks are not magnified). See also Supplementary Audios 1–6. **g** Principal component analysis (PCA) on acoustic parameters of the three simple sounds of SF and Pachón CF; variance ± SD for PC1 (5.7 ± 2.4) and PC2 (1.2 ± 1.1). **h** Confusion matrix (actual *versus* predicted) after pDFA (permutated discriminant function analysis) on the same dataset. Numbers indicate percentages of reclassification according to prediction (see color code). **i** Principal Component Analysis (PCA) on pulse rate parameters of 2 complex sounds of SF and Pachón CF. **j** The 3D plot on the right, including PC3, shows that the Serial Click and Serial Cloc clusters are well separated (variance ± SD for PC1: 2.34 ± 1.53; for PC2: 1.52 ± 1.23; for PC3: 0.73 ± 0.85). See also Supplementary Table 1 for summary of the statistics of PCA and pDFA analyses. Bdw, bandwidth; Fmin and Fmax, minimum and maximum frequency; Peak A , peak amplitude; Q25, 50, 75, quartiles 25, 50 and 75. Source data are provided as a Source Data file

**Table 1 Acoustic parameters for characterization of *A. mexicanus* repertoire**

| Parameters (means ± SEM) | Single cloc | | | Single click | | | Sharp click | | | KW; p; IMD |
|---|---|---|---|---|---|---|---|---|---|---|
| Morphotype | SF | CF | SF vs. CF | SF | CF | SF vs. CF | SF | CF | SF vs. CF | |
| n | 82 | 88 | | 77 | 89 | | 80 | 100 | | |
| Duration | 13 ± 0.6 | 17 ± 0.9 | 0.01 | 13 ± 0.9 | 11 ± 0.3 | ns | 12 ± 0.6 | 15 ± 0.7 | .001 | 59; 0.0001; b, d |
| Min–max (ms) | 10–30 | 10–50 | | 10–40 | 10–20 | | 10–50 | 10–30 | | |
| Peak frequency | 223 ± 4.0 | 201 ± 3.0 | ns | 3028 ± 296 | 2073 ± 236 | ns | 351 ± 18 | 339 ± 24 | ns | 289; 0.0001; a, b, c, d, e, f |
| Min– max (Hz) | 161–392 | 162–277 | | 185–7059 | 170–6895 | | 167–1156 | 162–1240 | | |
| Peak amplitude | −41 ± 0.8 | −36 ± 0.7 | 0.001 | −49 ± 0.9 | −51 ± 0.7 | ns | −41 ± 0.8 | −36 ± 0.7 | .01 | 213; 0.0001; a, b, c, d |
| Min–max (dB) | −55 to −22 | −52 to −19 | | −58 to −28 | −60 to −29 | | −57 to −28 | −53 to −18 | | |
| Min. frequency | 130 ± 1.0 | 125 ± 1.0 | 0.0001 | 205 ± 18 | 183 ± 14 | .0001 | 136 ± 2.1 | 137 ± 2.0 | ns | 103; 0.0001; a, b, c, f |
| Min–max (Hz) | 86–172 | 86–172 | | 43–947 | 120–1030 | | 129–215 | 86–215 | | |
| Max. frequency | 2481 ± 361 | 842 ± 133 | 0.01 | 20563 ± 403 | 17809 ± 629 | ns | 5501 ± 637 | 2896 ± 353 | ns | 380; 0.0001; a, b, c, d, e, f |
| Min–max (Hz) | 344–16193 | 300–10249 | | 2325–21963 | 2020–21963 | | 818–21963 | 387–14771 | | |
| Bandwidth | 2351 ± 361 | 712 ± 133 | 0.01 | 20358 ± 403 | 17621 ± 628 | ns | 5365 ± 637 | 2759 ± 353 | ns | 378; 0.0001; a, b, c, d, e, f |
| Min–max | 215–16063 | 170–10120 | | 2196–21920 | 1760–21834 | | 689–21834 | 215–14599 | | |
| Quartile 25 | 425 ± 34 | 275 ± 12 | 0.01 | 2881 ± 160 | 2313 ± 137 | ns | 626 ± 33 | 465 ± 25 | ns | 398; 0.0001; a, b, c, d, e, f |
| Min– max | 215–1550 | 210–1162 | | 1033–5943 | 430–6280 | | 215–1722 | 215–1205 | | |
| Quartile 50 | 1612 ± 150 | 872 ± 98 | 0.01 | 5426 ± 194 | 4452 ± 201 | ns | 1528 ± 97 | 1003 ± 55 | ns | 316; 0.0001; a, b, c, d |
| Min–max | 258–4651 | 250–4349 | | 1550–8785 | 1590–8828 | | 387–4737 | 344–2885 | | |
| Quartile 75 | 5678 ± 279 | 4504 ± 250 | ns | 10493 ± 143 | 9843 ± 162 | ns | 5454 ± 250 | 4237 ± 229 | ns | 318; 0.0001; a, b, c, d |
| Min–max | 344–9646 | 340–9560 | | 5641–12919 | 6546–13390 | | 1033–9474 | 473–9259 | | |

Mean acoustic parameters (±SEM, standard error on the mean) and minimum to maximum (min–max) indicating the range for each parameter are given for the three simple sounds. First quartile of energy spectrum, second quartile of energy (Q50), third quartile of energy (Q75) are given. Inter-morphotype differences for each type of sound are indicated in the 'SF vs. CF' columns. Values for Kruskal–Wallis (KW) tests comparing the three sound categories, in SF and CF, are presented in the last column and significant intra-morphotype differences (IMD) differences are indicated with lowercase letters: a, b = Single cloc vs. Single click, c, d = Single cloc vs. Sharp click and e, f = Single cloc vs. Sharp click. A Dunn's post hoc test showed significant (with p < 0.5, < 0.01, < 0.001, < 0.0001) or non-significant (ns) differences. CF = cavefish; Hz = hertz; ms = milliseconds; SF = surface fish

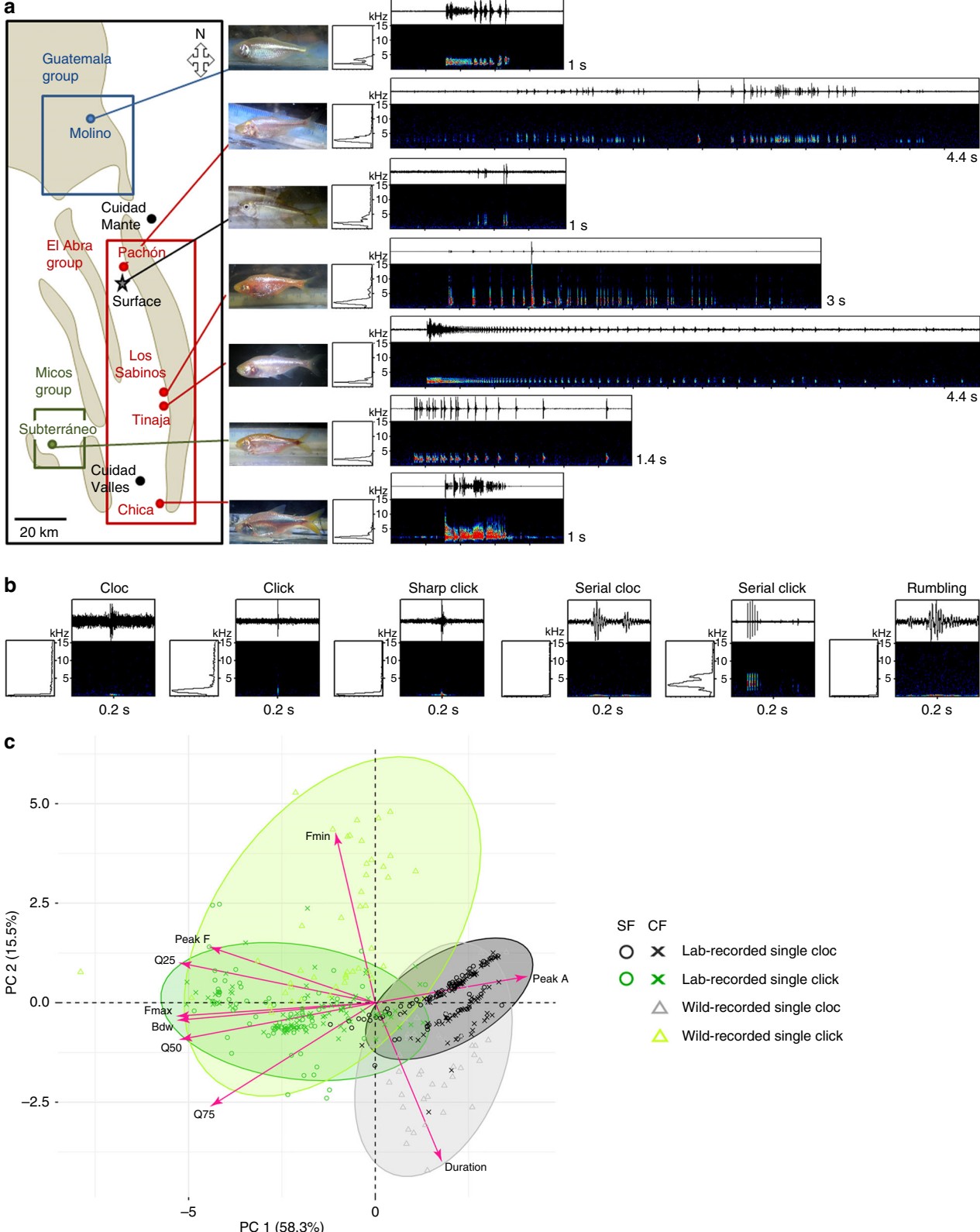

**Fig. 2** *A. mexicanus* is a sonic species in the wild. **a** Recordings in the wild. A schematic map shows the 3 proposed groups of independently evolved cave-dwelling *A. mexicanus* populations (blue, red, green rectangles). For each location sampled, a representative specimen photographed after recording (all pictures from authors), and a sonogram of a Serial Click-like sound are shown. See also Supplementary Audios 7–13. **b** Recordings from wild Pachón fish, showing the 6 types of sounds previously identified in laboratory conditions, here produced in the natural environment. See also Supplementary Audios 14–19. **c** Principal component analysis (PCA) on acoustic parameters of lab- and wild-recorded simple sounds of SF and Pachón CF (Variance ± SD for PC1 is 5.3 ± 2. and for PC2 is 1.4 ± 1.2). See also Supplementary Table 1 for the statistics of the PCA analysis. Bdw, bandwidth; Fmin and Fmax, minimum and maximum frequency; Peak A, peak amplitude; Q25, 50, 75, quartiles 25, 50 and 75. Source data are provided as a Source Data file

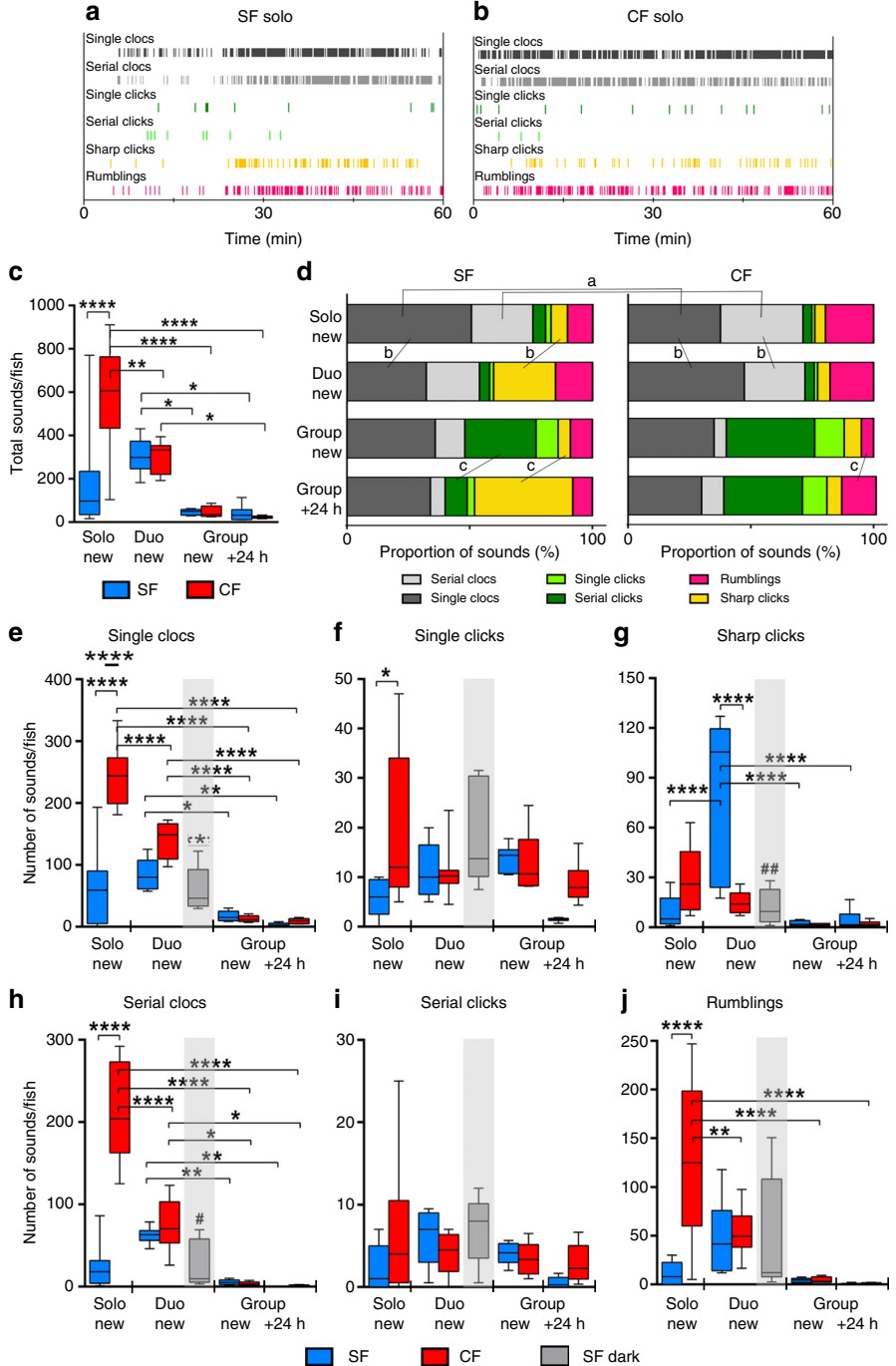

**Fig. 3** *A. mexicanus* sound production depends on contexts and morphotypes. **a**, **b**, Representative ethograms for a SF (**a**) and a Pachón CF (**b**) in solo condition, during one hour. Each vertical bar indicates an event and the color code for sounds is indicated. new: new environment; +24 h: after 24 h habituation. **c** Total numbers of sounds produced by SF (blue) and CF (red) in solo (**c**) and duo (**d**) context during 30 min (results are normalized *per* fish; 2-way ANOVA (interaction $F_{(3, 52)} = 9.1$; $p < 0.0001$). In this and the following figures, box plots show the distribution, median and extreme values (top and bottom whiskers) of samples. **d** Use of the repertoire. The proportions of sounds used by SF or CF, in solo or duo, are indicated in bars. Same color code for sounds as in ethograms. Fisher tests for comparisons: a indicates inter-morph significance between solo SF and solo CF; b indicates intra-morph significance between solo and duo context; c indicates intra-morph significance between new group and habituated group context. **e–j** Comparison of amounts of sounds produced during 30 min in solo, duo or group condition, in SF and Pachón CF, for the 6 sound categories (results are normalized *per* fish; 2-way ANOVA interactions: $F_{(2, 38)} = 20.2$; $p < 0.0001$ (**e**), $F_{(2, 38)} = 3.8$; $p = 0.03$ (**f**), $F_{(2, 38)} = 17.7$; $p < 0.0001$ (**g**), $F_{(2, 38)} = 41.6$; $p < 0.0001$ (**h**), $F_{(2, 38)} = 1.6$; $p = 0.2$ (**i**), $F_{(2, 38)} = 9.3$; $p = 0.0005$ (**j**)). On each graph, the sound type, the morph (SF in blue, Pachón CF in red), and the context are indicated. Gray bars correspond to SF duos in the dark. Bonferroni *posthoc*: ****$p < 0.0001$, **$p < 0.01$, *$p < 0.05$. Duo SF in the light and in the dark were compared separately with Mann–Whitney tests: $U = 3$; $p = 0.005$ (**g**), $U = 7$; $p = 0.03$ (**h**), #$p > 0.05$, ##$p < 0.01$. Data are means ± SEM. See also Supplementary Data 3 for exhaustive statistics. Source data are provided as a Source Data file

sounds, both when discovering the new environment and after habituation when the social group was formed. A two-way ANOVA analysis on these data indicated a significant interaction between the two independent variables (morph and social context) on the dependent variable, sound production ($p < 0.002$, Bonferroni *posthoc*). Thus, the effect of social context on the quantity of sounds produced is dependent on the morph.

Further, specific patterns of variations according to social context were observed for each sound type in each morph (Fig. 3e–j). Clocs, Serial Clocs and Rumblings were globally decreased when group size increased from 1 to 2 to 6 fish, especially for Pachón CF (Fig. 3e, h, j). On the contrary, Clicks and Serial Clicks were largely unaffected by group size in either morph (Fig. 3f, i). Noteworthy, Sharp Clicks production was particularly substantial in two conditions: SF duos and SF groups after habituation (Fig. 3g).

The use of the sound repertoire in SF and Pachón CF was also compared in the solo/duo/group conditions (Fig. 3d). Clocs and Serial Clocs were the most commonly produced sounds in the two morphs (grey shades on Fig. 3d). Overall, the proportions of sounds used varied between morphs in a given context, and between contexts in a given morph (Fisher tests). The most striking variations were a significant use of Sharp Clicks in SF duos and SF groups after habituation (yellow on Fig. 3d), and a more prominent use of Clicks in groups (green on Fig. 3d). In summary, these data showed that sound production is quantitatively and qualitatively variable according to behavioral contexts and fish morphs, suggesting that the sound repertoire is used to convey information and that this use is changed in cavefish.

**Sound production and vision**. To investigate the role of vision in sound production, we repeated the SF duo experiments in the dark (Fig. 3e–j, grey bars). Absence of light had very different effects on the 6 sound types. The emission of Clocs (Single and Serial) was decreased by about 50% when compared to SF duos in the light. Clicks (Single and Serial) and Rumblings were unaffected by the lack of vision. And Sharp Clicks were very strongly reduced in the dark, reaching the scores of Pachón CF duos (Fig. 3g). These results suggested that the production of Sharp Clicks (and Clocs to a lesser extent) is visually-triggered in SF, while that of Clicks and Rumblings is independent of vision. Overall, the variations in the production of sounds observed in the different social contexts and light conditions led us to pose a hypothesis on the meaning and significance of sounds produced by *A. mexicanus*. Below we focused on the behavioral relevance of Sharp Clicks, and the other sounds will require further studies.

**Sharp Clicks, aggressiveness and feeding behavior**. Sharp Clicks were over-represented in SF duos and habituated SF groups, with a proposed visual trigger. High levels of aggressiveness were observed on the corresponding videos, suggesting that Sharp Clicks may be used in SF agonistic behavior. To test this hypothesis, we performed resident-intruder assays, an appropriate way to compare aggressiveness in eyed and eyeless animals because it does not require vision[16,28]. In brief, individual fish were habituated in a tank overnight, and the next day one fish (the intruder) was transferred in the tank of the other (the resident). As expected, attacks, chasing and obstructing events were numerous in SF but not in CF (Fig. 4a, c), and they represented the establishment of hierarchy between the dominant and the subordinate[16]. Such aggressive behavior in SF was paralleled by an abundance of Sharp Clicks and Rumblings, almost absent in CF in the same test (Fig. 4d). A correlation analysis performed on one second bins during the one-hour test further showed good correlation coefficients in time between Sharp Clicks and

intensive aggressiveness in SF but not in CF (Fig. 4b; Pearson's coefficient 0.46; see also Supplementary Table 2), and multiple observations suggested that Sharp Clicks were emitted by the attacker/dominant fish, during the strike at the subordinate (Supplementary Movie 1). This notion was supported by the positive relationship with excellent regression coefficients between attacks or chases and the number of Sharp Clicks emitted (Supplementary Fig. 3; $r^2 = 0.87$ and $r^2 = 0.84$, respectively, $p < 0.05$). Of note, in the resident-intruder assay Clocs were not correlated to attacks or chases, serving as controls and strengthening the specificity of the Sharp Click/attack association. These observations also supported the idea that contrarily to Sharp Clicks, Clocs were not exclusively produced by the dominant fish. The negative relationship between Clocs and obstruction events further suggested that Clocs were emitted by the subordinate fish (Supplementary Fig. 3).

To substantiate these findings, aggressive behavior in SF was also tested using a classical mirror test. A high number of Sharp Clicks was produced while the fish repeatedly attacked its own image in the mirror, with good temporal correlation on ethograms (Fig. 4e). A transition frequency analysis showed that in most cases, attacks were just preceded or just followed by events of Sharp Clicks, Rumblings, and a position close to the mirror (Fig. 4f). A correlation analysis performed on one second bins during the whole test further showed excellent correlation in time between occurrences of attacks and the production of Sharps Clicks and Rumblings (Fig. 4g; Pearson's coefficients are 0.64 and 0.46 respectively; see also Supplementary Table 3). This supported the idea that the vast majority of Sharp Clicks must be produced by the attacker/dominant fish in the resident-intruder assay. These interpretations also fit with the abundance of Sharp Clicks in well-established social groups (with established hierarchy and territoriality supervised by the dominant individual) but not in new groups (phase of observation between the 6 tank mates) (Fig. 3d). Accordingly, we propose that in SF, Sharp Clicks are visually-triggered intimidating sounds produced by dominant fish during the establishment and the maintenance of hierarchy, and they are associated to aggressive behavior. Of note, both the resident-intruder and the mirror assay revealed that Rumblings were also strongly associated with the sequences of aggressive behaviors.

On the other hand, blind Pachón cavefish did also produce Sharp Clicks (Fig. 3g). However, (1) they were not visually-triggered, (2) they were intriguingly most produced in solo, which seems incompatible with a social use, and (3) they probably did not have an agonistic value (no relationship between Sharp Click and attack numbers in the resident-intruder assay). Therefore, the trigger, the use and the meaning of Sharp Clicks has changed in cavefish. We hypothesized that this change may have accompanied the proposed cavefish behavioral shift from fighting to foraging[16,31]. To explore the use of Sharp Clicks during food-seeking behavior in CF, recordings were performed in starved, overfed and control Pachón fish before and after perfusion of an odorant food solution (crushed and filtered granular fish food, see Methods) that triggers foraging activity and its associated cavefish typical posture[32], especially in starved animals (Fig. 4h and Supplementary Fig. 1c). Sharp clicks (but not the 5 other sounds; Single Clocs are shown as controls) were specifically and strongly produced by starved cavefish after chemosensory stimulation, in parallel to intense foraging activity on the bottom of the tank and an increase in swimming speed shortly after odor presentation (Fig. 4h, i, l and Supplementary Fig. 4a). Likewise, a positive relationship was observed between the number of Sharp Clicks (but not Clocs or Rumblings), and the time spent foraging (Fig. 4m and Supplementary Fig. 4c). These data suggest that in cavefish, Sharp Clicks are foraging-related and chemosensory-

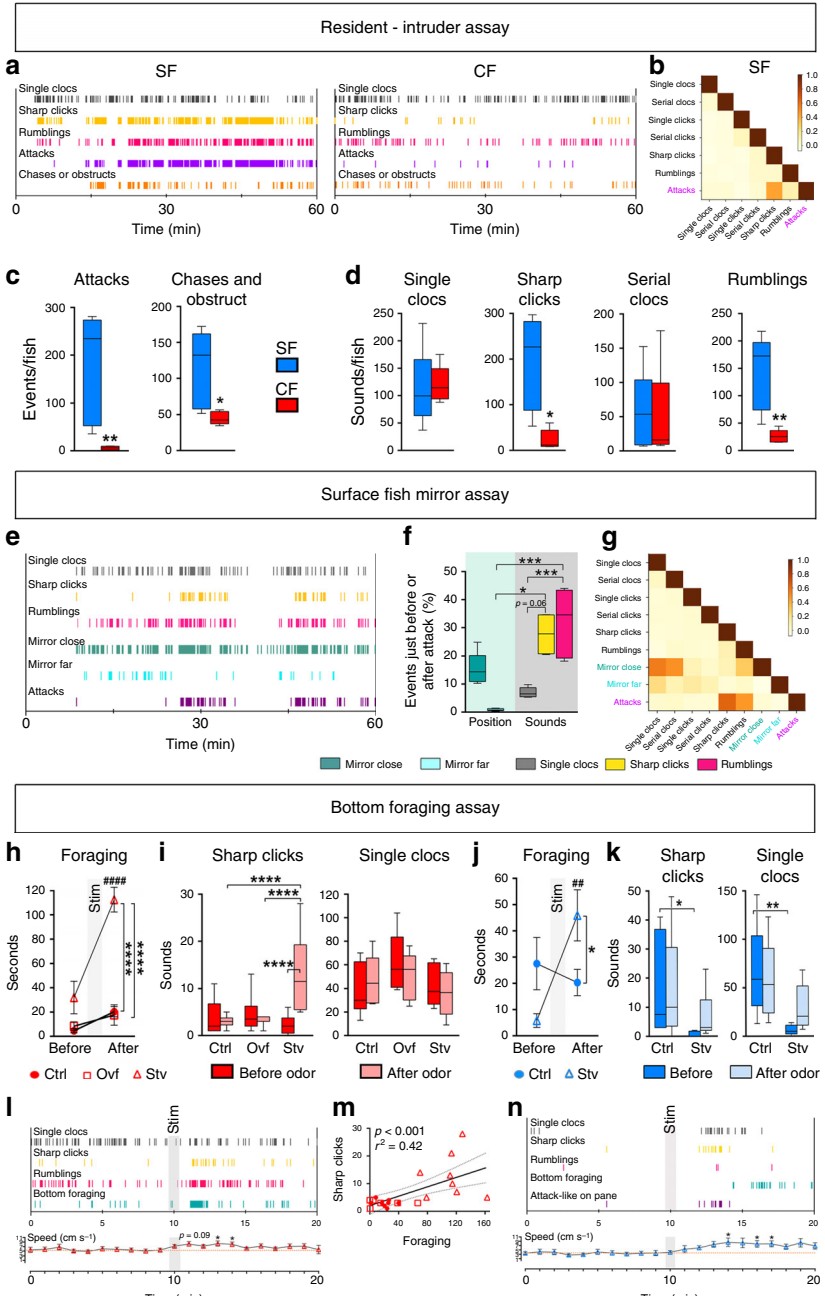

**Fig. 4** Comparison of Sharp Clicks use and significance in cavefish and surface fish. **a** Representative ethograms for SF and Pachón CF during a one-hour resident-intruder assay. Each vertical bar indicates an event and the color code is indicated. **b** Heat map showing the correlation for SF between the occurrence of two events at one second bin level throughout the one-hour test (Pearson's coefficients are color coded from beige to dark: $0 \leq r \geq 1$; See also Supplementary Table 2). **c, d** Comparison of amounts of agonistic behavioral events and sounds produced during a one-hour resident-intruder assay, normalized *per* fish. On each graph, the behavior, the sound type, and the morph (SF in blue, Pachón CF in red) are indicated. Mann–Whitney tests for attacks and chases (**c**), respectively, $U = 0$ and $U = 2$, and Sharp Click and Rumbling (**d**), respectively, $U = 1$ and $U = 0$ (*$p < 0.05$, **$p < 0.01$). **e** Representative ethogram of a mirror assay in SF. **f** Transition frequency analysis showing the events that just preceded or just followed an attack, in percent, throughout the one-hour test, showing the most frequent associations and sequences of events. Kruskal–Wallis 25.2; $p < 0.0001$. **g** Heat map showing the correlation between the occurrence of two events at one second bin level throughout the one-hour test (Pearson's coefficients are color coded from beige to dark: $0 \leq r \geq 1$; See also Supplementary Table 3). **h–k**, Comparison of amounts of foraging events and sounds produced by fish in the 10 min before and after perfusion of an odorant food solution. Pachón CF in red, Surface fish in blue. Ctrl, control; Ovf, overfed; Stv, starved. Two-way ANOVA-RM, interactions: $F_{(2,28)} = 11.20$; $p = 0.0003$ (**h**), $F_{(2,21)} = 13.32$ (**i**, Sharp Clicks), $F_{(1,16)} = 12.42$; $p = 0.003$ (**j**), $F_{(1,16)} = 2.01$; $p = 0.176$ (**k**, Sharp Clicks), $F_{(1,16)} = 2.5$; $p = 0.13$ (**k**, Single Clocs). Dunn's *posthoc*: *$p < 0.05$, **$p < 0.01$, ****$p < 0.0001$ inter-diet conditions and ##$p < 0.01$, ###$p < 0.001$ within diet condition in foraging. **l, n** Representative ethograms of a starved CF (**l**) and a starved SF (**n**) and respective speed variations of all tested individuals (Friedman $p = 0.03$, *$p < 0.05$; Dunn's *posthoc* $p < 0.05$ calculated from baseline during 10 min) during foraging assay. The grey shading indicates the period of odor perfusion. **m** Linear regression showing the relationship between sound production and time spent in bottom foraging behavior in CF ($F = 15.65$). Data are mean ± SEM. See also Supplementary Data 3 for exhaustive statistics. Source data are provided as a Source Data file

triggered sounds, and they are especially used in conditions of low metabolic reserves, in starved conditions. Finally, to ascertain the change in Sharp Click-related behavior in CF, the foraging test was also performed on SF. Like in CF, starved SF increased bottom foraging behavior in response to food odor (Fig. 4j), but in a delayed manner and without temporal nor quantitative correlation with Sharp Clicks production (Fig. 4j, k, n and Supplementary Fig. 4b, d). Instead, Sharp Clicks produced by starved SF shortly after chemosensory stimulation were simultaneously produced with attack-like behavior against the aquarium window (Fig. 4n), again highlighting the association between Sharp Clicks and agonistic behavior in the surface morph. Thus, while extant cavefish and surface fish still share their acoustic repertoire (and they must also share the mechanical apparatuses to produce these sounds), they diverge in their use of acoustic signals along with their behavioral specializations in their specific habitats[12,33].

Finally, in order to establish the meaning of Sharp Clicks as true communicative sounds, we performed play-back experiments. Groups of 6 SF and 6 CF were exposed to audio bands playing Sharp Clicks (originating from the same morph) or white noise (as control) and their behavior was analyzed (Fig. 5a, b). SF transiently stopped agonistic behavior (Fig. 5c) and gathered (Fig. 5d, e) upon hearing the two types of sounds, showing that they responded to the playbacks. Only the white noise elicited an increase in swimming speed, suggesting that this stimulus was stressful, as previously reported for example in goldfish[34], and as opposed to the Sharp Clicks which are familiar sounds (Fig. 5f). Accordingly, the dominant fish occupying zone O (Opposed to the speaker; Fig. 5a) moved to other zones to join the rest of the school during white noise stimulation while this displacement was less significant during Sharp Clicks play-back, and without increasing speed (Fig. 5g, h and Supplementary Movies 2 and 3). We interpret these data as representing a transient abolition of hierarchy in the group, induced by Sharp Clicks. Cavefish reactions to play-backs were markedly different. They hardly showed any response to the two types of sounds in terms of group structure (Fig. 5j, k) but they increased their swimming speed in a delayed manner after white noise stimulation (Fig. 5l). Strikingly, CF which had a tendency to naturally occupy the zones O and M (far from speaker; Fig. 5b), changed their position to zone S (Speaker zone) during Sharp Clicks stimulation, whereas white noise did not elicit any change of place in the aquarium (Fig. 5m, n and Supplementary Movies 4 and 5). This is consistent with the Sharp Clicks "feeding signal" triggering an attraction of CF towards the zone of emission of the sound. Of note, such a rapid attraction of cavefish towards another cavefish searching for food has been described by Hüppop[35]. The underlying mechanism may be the acoustic communication signal described in the present study. Altogether, these data indicate that both SF and CF show behavioral responses to sounds produced by conspecifics of their own morphotype, demonstrating a case of real acoustic communication; and that the nature of the behavioral response to a given sound is markedly different in the two morphs, highlighting the evolution or shift of this acoustic communication, accompanying the changes in their behaviors.

## Discussion

*A. mexicanus* appears to be a powerful model to address the question of the role of acoustic communication in evolution, and possibly further, in speciation, as proposed for example between the tilapia sister species *Oreochromis niloticus* and *Oreochromis mossambicus*[8]. In this respect, we have recently reported that cave and surface *Astyanax* breeding behavior is identical and occurs during the night, which allows natural cross-breeding between the two morphs[36]. We have hypothesized that acoustic signals may be involved, in the absence of visual modality, for reciprocal interactions between the male and the female during reproduction. This idea is now supported by the present finding that *A. mexicanus* is a sonic species, and deserves further studies; but we can predict that contrarily to agonistic or foraging behaviors, the acoustic signals associated to reproductive behavior should not have changed yet between the two morphs.

By contrast to their degenerated visual system[37–41], the olfactory[21,42], the gustatory[22,23,43], and the mechano-sensory lateral line[24,44] systems of cavefish have evolved constructively, presumably as adaptive compensatory mechanisms for the loss of vision in the dark cave habitat. Their auditory system seems neither enhanced nor degenerated, as SF and CF hearing capacities in terms of threshold sensitivity and bandwidth do not differ[25], their ears seem to develop similarly[21], and the morphology of their labyrinths[45] and Weberian ossicles[46] do not specifically differ. The reasons for the maintenance of hearing in cavefish have been considered enigmatic, raising the question of 'what they are listening to' in their caves[25,47]. Indeed, food or water droppings on the water surface are rather in the neuromast frequency detection range (10–150 Hz[24,48,49]); bats sounds are probably too high in frequency (although high frequency hearing is reported for another family of teleosts, the clupeids, which possess a specialized organ called the utricle in their inner ear;[50,51]); and sound-producing arthropods such as crayfish[52] are occasionally found only in a few caves (but not in the Pachón cave). The present work now brings an answer to this puzzling question of what they are listening to: cavefish listen to their conspecifics.

## Methods

**Fish samples.** Laboratory stocks of *A. mexicanus* surface fish (origin: San Salomon spring, Texas, USA) and cavefish (Pachón population) were obtained in 2004 from the Jeffery laboratory at the University of Maryland, College Park, MD, USA. The colonies were maintained at 23 °C (cavefish) or 26 °C (surface fish) on a 12:12 h light:dark cycle. Fish care, feeding, growth conditions and breeding are described in ref. [53]. Animals were treated according to the French and European regulations for handling of animals in research, and we have complied with all relevant ethical regulations for animal testing and research. SR's authorization for use of *Astyanax mexicanus* in research is 91-116 and the Paris Centre-Sud Ethic Committee protocol authorization number related to this work is 2012-0054. The animal facility of the Institute received authorization 91272105 from the Veterinary Services of Essonne, France, in 2015. Adult male and female fish aged between 1 and 8 years old (Pachón and surface fish), sized between 3.5 and 6.5 cm, and born in our facility were used.

Field recordings were obtained during two field expeditions in the states of San Luis Potosi and Tamaulipas, Mexico, in March 2016 and March 2017, under the auspices of the field permit 02438/16, delivered by the Mexican Secretaria de Medio Ambiente y Recursos Naturales. We recorded from 6 caves hosting *Astyanax mexicanus* troglomorphic cavefish populations (map on Fig. 1). We also sampled *Astyanax mexicanus* surface fish in a well located in the village of Praxedis Guerrero.

**Sound recordings and analyses.** Laboratory recordings were performed in 7 L tanks for solo (*n* = 10 tests/morph), duo (*n* = 8 tests/morph in the light and *n* = 6 tests for SF in the dark; hence twice the number of fish were used), mirror tests (*n* = 6 tests with SF), resident-intruder assays (*n* = 10 tests/morph; hence twice the number of fish were used) and foraging assays (*n* = 8 tests for each CF condition: control, starved, overfed; *n* = 10 or 8 tests for each SF condition: control or starved, respectively). Group assays (*n* = 6 tests/morph, with 6 fish/test) were performed in 25 L tanks. For solo, duo and group analyses, sound production was compared during the first 30 min of discovery of a new environment and new conspecific(s), or after 24 h habituation. Recordings were obtained in soundproof rooms insulating from outside noise. Inner wall-glasses of the tanks were covered with foam to prevent inside echoes (Supplementary Fig. 1a). Sounds were recorded using a hydrophone (H2a-XLR, Aquarian Audio products, Anacortes, WA, USA; sensitivity: −180 dB, re 1 V/µPa, flat frequency response: ±4 dB, 20 Hz–4.5 kHz) connected to a pre-amplifier (Yamaha MLA8 for solos and duos as in refs. [6,54], RME OctaMic II for other conditions). Sounds were synchronized with images provided by video cameras (Active Media Concept BUL520 for solos and duos, high definition Grundig GCH-K1305B-1 for other conditions) using a specific video card (Osprey 450e for solos and duos, Blackmagic Decklink 4K for other conditions).

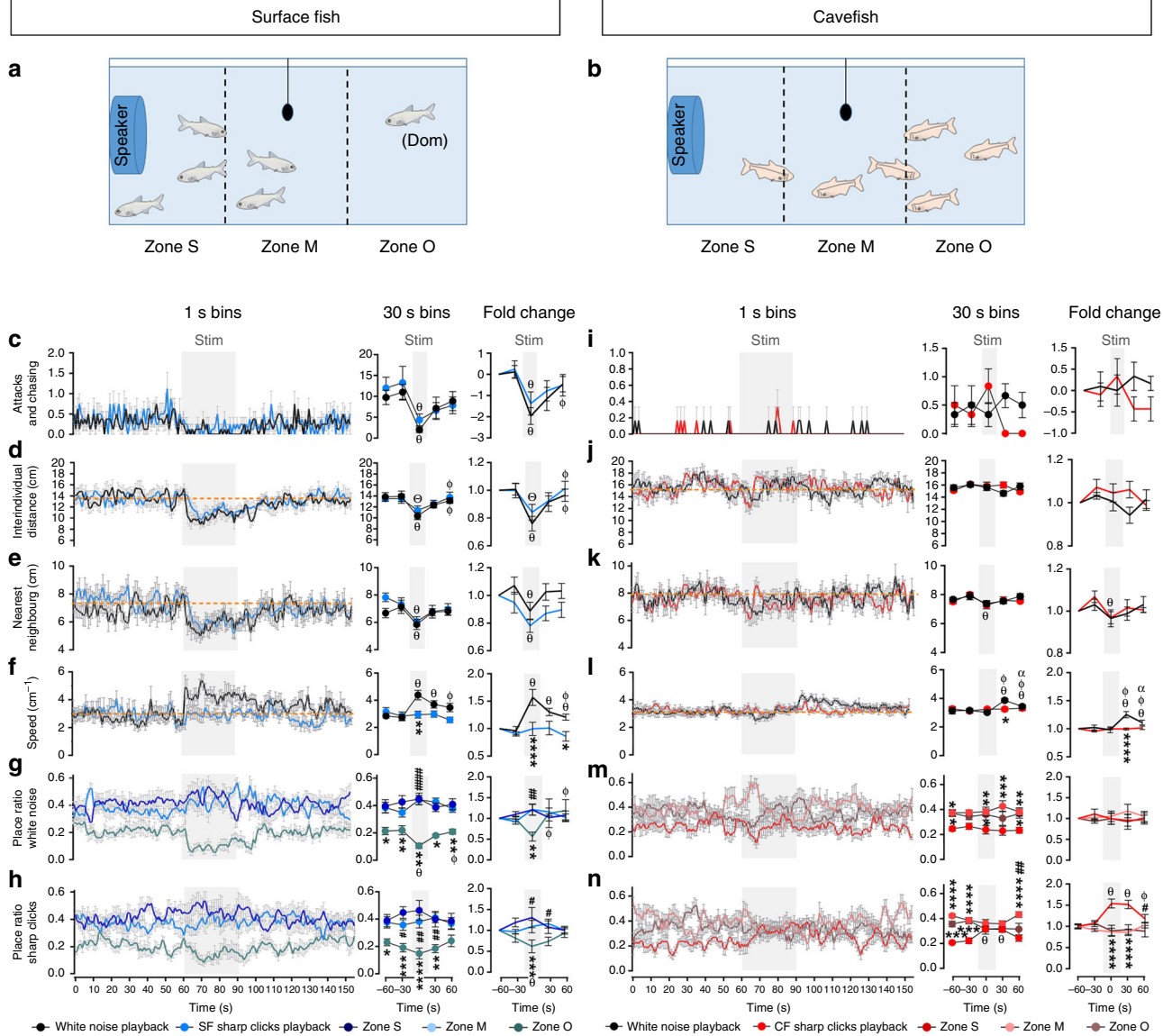

**Fig. 5** Effect of Sharp Clicks or white noise play-back on fish behavior. Several behavioral parameters were measured before, during and after play-back (grey area, Stim) of Sharp Clicks (colored lines) or white noise (black lines) in surface fish (blue graphs, left columns) and cavefish (red graphs, right columns). Results are presented at one second bins and 30 s bins, as well as in fold change respective to the "before sound play-back" condition. **a**, **b**, Schemes of the experimental set-up, with representative distributions of the fish in the 3 zones of the tank, respective to the speaker (fish drawings are from authors). dom, dominant SF individual. See also Supplementary Movies 2-4. **c**, **i**, Agonistic behavior. Two-way ANOVA-RM, interactions: $F_{(4,56)} = 0.42$; $p = 0.80$ (**c**), $F_{(4,40)} = 2.18$; $p = 0.08$ (**i**). **d**, **j**, and **e**, **k**, average inter-individual distance and nearest neighbor distance, respectively. Two-way ANOVA-RM, interactions: $F_{(4,56)} = 0.66$; $p = 0.62$ (**d**), $F_{(4,40)} = 2.66$; $p = 0.05$ (**j**), $F_{(4,56)} = 1.36$; $p = 0.26$ (**e**), $F_{(4,40)} = 0.52$; $p = 0.72$ (**k**). **f**, **l** swimming speed. Two-way ANOVA-RM, interactions: $F_{(4,56)} = 6.004$; $p = 0.0004$ (**f**), $F_{(4,40)} = 11.3$; $p < 0.0001$ (**l**). **g**, **m**, and **h**, **n**, place preference index in the 3 zones of the aquarium (S, M, O, see color code) during white noise or Sharp Clicks play-back, respectively. Two-way ANOVA-RM, interactions: $F_{(8,84)} = 3.9$; $p = 0.0006$ (**g**), $F_{(8,84)} = 2.08$; $p = 0.047$ (**h**), $F_{(8,60)} = 0.73$; $p = 0.66$ (**m**), $F_{(8,60)} = 5.06$; $p < 0.0001$ (**n**); Bonferroni posthocs: $p$-values $< 0.05$–$0.0001$ between time points are indicated as compared to baseline before stimulation (θ), to the stimulation period (Φ), to 30 s post stimulation (α), and Θ indicates trends as compared to before stimulation ($p \leq 0.08$); $p$-values $< 0.05$–$0.0001$ at time points between white noise versus sharp clicks playbacks (*), from zone S (*) and from zone O (#). Stim, play-back stimulation; zone S, M, O, close to speaker, median zone, opposed to the speaker, respectively. Data are means or fold changes ± SEM. See also Supplementary Data 3 for exhaustive statistics. Source data are provided as a Source Data file

The distance between hydrophone and recorded fish was under the attenuation distance, estimated in each type of tank using specific equations (Akamatsu, 2002). Additional infrared cameras placed above tanks allowed individual and group tracking to measure speeds, nearest neighbor distances, inter-individual distances and fish positions in tanks (Viewpoint behavior technology, Civrieux, France).

For natural field recordings, CF were directly recorded in their hosting pools in 6 different caves, in the dark, either from 10–12 fish inside a large net installed in their natural pool, or from freely swimming fish in the case of small natural pools. Cavefish ($n = 8$/cave) were also systematically recorded in 40 L plastic pools installed and left on site overnight, except in the Molino cave, to ascertain that

recorded sounds were only produced by CF. SF were recorded in the light, in the still water of a well (Supplementary Fig. 1b). Hydrophones were connected to portative pre-amplifiers (ART Dual Pre USB, NY 14305, USA) and recorders (Zoom H4n, NY 11788, USA) with SD cards, and recording parameters were adjusted with direct audio listening depending on environmental acoustic characteristics of each cave. They were left on sites for overnight recordings.

Sounds were extracted from audio-video recordings by ear. They were digitized at 44.1 kHz (16-bit resolution) and analyzed using fast Fourier transform (FFT) with Avisoft SAS Lab Pro 5.2.07 software (Avisoft bioacoustics, Glienicke, Germany)[6]. The acoustic structure of simple sounds was characterized in two ways.

Lab- and wild-recorded simple sounds (Single Clocs, Single Clicks and Sharp Clicks) were analyzed using a set of 1 temporal parameter, the duration, measured from the oscillograms, and 8 spectral parameters obtained from power spectra (FFT, window type: Hann, window size: 512; time overlap: 90%) within a 0–22.5 kHz bandwidth: peak frequency (mode) of the frequency spectrum, amplitude at the peak frequency, first quartile of energy (Q25), i.e. the frequency value corresponding to 25% of the total energy spectrum, second quartile of energy (Q50), third quartile of energy (Q75), minimal and maximal frequencies of the spectrum, and the bandwidth, i.e. difference between maximal and minimal frequencies. Complex sounds (Serial Clocs and Serial Clicks) were examined using 5 fine temporal parameters including sound duration, pulse number, mean inter-pulse duration, mean pulse duration, and pulse rate (= pulse number/sound duration) (R package Seewave). Pulses were considered "simple" if they were of short duration (< 20 ms) and separated by > 1 s interval from the next pulse (threshold defined from the histogram of the inter-pulse durations). Comparisons between the two morphotypes and between recording conditions were performed using principal component analysis (PCA; R package FactoMineR) and permutated discriminant analysis (pDFA; R routine from Bertucci et al. 2010). Sounds used for these analyses were high-pass filtered at 150 Hz. For each experiment/recording, sounds and behaviors were systematically scored for quantitative comparisons and to generate ethograms (ODREC5, Observational data recording software, S. Péan, IFREMER, France). All supplementary audios provided might be listened to with headphones.

**Aggressive behavior tests**. The tests were performed in a room at 25 °C, in lighted conditions or in the dark, as indicated. All fish were tested once, and there was no difference in the behavior according to sex.

For the resident-intruder assays, fish (SF or CF) were isolated overnight in a 7 L tank. The next morning, an 'intruder' was transferred into the tank of a 'resident' fish and interactions between them were recorded during one hour using a camera (front camera—high definition Grundig GCH-K1305B-1, and top infrared camera —Viewpoint), while recording sounds synchronously. In a given test, the 2 fish were always of the same morph and similar size. An attack was defined by the charge of a fish and the escape of the other. An obstruction event was defined as a fish blocking or obstructing the other in a corner or restricted part of the aquarium, or preventing it from escaping. In text and Fig. 4, the n numbers given indicate the number of tests (hence the number of animals used is 2n).

For the mirror tests, surface fish were individually transferred into a 7 L tank in which one side was covered by a mirror. Aggressive attempts against its own image in the mirror were immediately recorded synchronously with the sounds, and during one hour.

**Foraging behavior tests**. Pachón cavefish were either starved (no food), overfed (double dose) or fed normally during 3 months. The day of the test, they were habituated in solo for 20 min in a 7 L tank, and synchronized behavioral and acoustic recordings were obtained during the next 20 min: 10 min before and 10 min after the delivery at the surface of the water of 1 ml of a solution of crushed and filtered granular fish food (5 g for 50 ml solution; TetraDiskus; Tetra, Blacksburg, VA, USA), as in[42]. For analysis, the swimming speed, as well as the position of the fish in the water column and its posture head-down swimming on the bottom of the aquarium were scored (Supplementary Fig. 1c). Control and starved surface fish were assayed following the same procedure.

**Play-back experiments**. Groups of 6 individuals (8 surface fish groups and 6 cavefish groups) underwent 24 h of habituation in a 25 L tanks with a speaker turned on (UW30DLREV under water speaker, Lubell LABS, Whitehall OH, USA), connected to an amplifier (ATOLL 50SE, ATOLL Electronique, Brecey, France). Sound tracks of 30 min were designed with alternate sequences of 30 s of white noise, used as a control sound, and 30 s of repeated Sharp Clicks extracted from audio bands recorded during SF or CF resident-intruder assays. Each sound was played twice in a randomized manner, i.e. soundtracks started either by white noise or Sharp Clicks and stimulations were separated by 5 min of intercalated silences. Sharp Clicks playback levels were adjusted according to the Sharp Click decibel range emitted by each morph. Behaviors and sounds were manually analyzed at one second bins for comparisons between the 60 s before, 30 s during (except for sounds, as they were not detected during playback sequences) and 60 s after playback stimulations. Behaviors (group structure, swimming speed, position in the tank) were analyzed with Viewpoint software.

**Statistics**. Normality was assessed with a Kolmogorov–Smirnov. Mann–Whitney two-tailed, Kruskal–Wallis or Friedman (for repeated measures) tests followed with Dunn's post hocs were performed on non-normally distributed data sets. Two way-ANOVAs followed with Bonferroni post hoc tests were performed to compare sound production in SF and CF in solos, duos and groups under light condition (Fig. 3). Mann–Whitney tests were used to compare the quantity of sound produced by the two morphs, the number of aggressive events in light and dark SF duos (Fig. 3) as well as in resident-intruder and mirror assays (Fig. 4). Transition frequencies and temporal correlation (coefficients were calculated with Pearson's correlation) are shown *per* one second bins in the resident-intruder and mirror

assays. Two-way ANOVAs with repeated measures (Two-way ANOVA-RM) were used to compare variations in the foraging assay before and after food odor stimulation (Fig. 4) and in playback experiment (Fig. 5). In figures, box plots show the distribution, median and extreme values (top and bottom whiskers) of samples. Slopes in linear regressions show relationships between sounds and behaviors in resident-intruder assay and foraging assay. Sound proportions in Solo, Duo and Groups were analyzed using Fisher exact test comparing the quantity of one sound versus the quantity of all the others sounds in SF and CF (Fig. 3). Delta (Supplementary Fig. 5) or fold change were calculated to highlight differences or variations, and $\log_2(\text{fold change} + 1)$ transformation was applied on data sets containing values = 0 (attacks and chases in Fig. 5). Statsoft Statistica 6, GraphPad Prism 6, FactoMineR library from R 3.1.3[55] and Matplotlib from Python 3.6 softwares were used for statistical analyses and graphical representations. Supplementary Data 3 provides all statistical values for all tests performed in this paper.

**Reporting summary**. Further information on research design is available in the Nature Research Reporting Summary linked to this article.

## Data availability
All data generated or analyzed during this study are included in this published article (and its Supplementary information files). The source data underlying Figs. 1g–i, 2c, 3c, 3e–j, 4c, d, f, 4h–n, 5c–n, Supplementary Fig. 2, Supplementary Fig. 3, Supplementary Fig. 4, and Supplementary Fig. 5 are provided as a Source Data file.

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

## Acknowledgements

This work was supported by a Lidex Neuro-Saclay collaborative grant to S.R. and J.A., an Equipe FRM grant (DEQ20150331745) to S.R., a prize from Fondation des Treilles to C. H., and an Ecos-Nord exchange Program to S.R. and Patricia Ornelas-Garcia. We thank Luis Espinasa, Julien Fumey, Didier Casane, Stéphane Père and all other members of the Rétaux lab for their help in data collection in the field, and Patricia Ornelas-Garcia for obtaining a shared field work permit. We thank Laura Chabrolles from ENES lab for her help detecting first sounds of *Astyanax mexicanus*. This work benefited from the valuable help of Cynthia Froc at the Amatrace platform for the analysis of fish behavior. We are grateful to Michel Engeln (UMB, Baltimore, MD, USA) for fruitful discussions on behavior, and to Thomas Deneux (Neuro-PSI, Gif-sur-Yvette, France) and Mickael Orgeur (Institut Pasteur, Paris, France) for valuable inputs in digital artwork.

## Author contributions

S.R., C.H., and J.A. designed the study, C.H. performed all the experiments, C.H., S.R., and J.A. analyzed the data, and S.R. wrote the paper. All authors discussed the results and commented on the paper.

## Additional information

**Competing interests:** The authors declare no competing interests.

