## [Peer Review File · Nature Communications]

Reviewers' Comments:

Reviewer #1:

Remarks to the Author:

The manuscript documents the exciting discovery of sound production in the surface and cave forms of the Mexican cave tetra, *Astyanax mexicanus*. This discovery further enriches our understanding of the sensory biology of this species, which has become a model species for genetics/genomics of development and sensory biology.

The work is described well, and the design of the work is very good. I cannot comment on the statistical analysis. Ability to reproduce work should be straightforward.

Specific issues - pertaining to English word usage (for accuracy) and some scientific issues.

Line 25 - should be "allows the exchange of information..."

Line 31 - chemosensory and mechanosensory should not be hyphenated

Line 32 - "probably"?? - there is data in the literature.

Line 33 - citation 17 - this paper is from 1970 - is there more recent data?

Line 34 - how is "evolution" being defined in this paper? If the paper is only comparing surface vs. cavefishes, is "evolution" the best term? Is it a "shift"? What term was used in the papers about the lateral line-mediated VAB? See throughout summary and on line 58 as well.

Line 36 - you didn't just "find" something - this is an important "discovery" - use that term.

Line 39 - is "morphotype" used to define the surface and cave "types", but please use the same term throughout the manuscript.

Line 50 - perhaps say "...widely used among terrestrial and aquatic animals for..."

Line 51 - should be "teleost fishes" - check throughout.

Line 53 - should be "behaviors are starting to be understood"

Line 54 - should be "evolve via adaptation within species with respect to needs in their specific..."

Line 56/57 - should be "which diverged.....experience markedly....."

Line 63 - say "First we determined...". Then start next sentence with "Six different types....including 3 simple sounds and 3 complex sounds were recorded from 60 hours....."

Line 72 - the cited study used a behavioral assay - this should be mentioned.

Line 72/73 - This point needs to be made after the different sounds are defined...in the results.

Line 75 - what is meant by "to validate human hear identification"?

Line 84 - Is "or" correct?

Line 86 - should be "caves with...."; also in line 91 - should be "in a well in which surface forms are found".

Line 94 - What "repertoire" is being referred to here?

Line 96 - should be "pursued additional analysis of sounds in...."

Line 101 - elaborate on the implications of absence of a visual modality, e.g., "causing them to be more reliant on other sensory systems".

Line 101-102 - Evolution of acoustic communication in cavefish??? Or are the differences between surface fish and cavefish the focus? This is confusing.

Line 106 - should be "used for acoustic communication..., then sound production should vary...."

Line 108 - instead of "solo", "duo" which is awkward, please use "individual fish", "pairs of fish", "groups of 6 fish". Check this throughout.

Line 108 - I think that "discovery" should be "exploration", which is a term used in the literature.

Line 121 - change "morph-dependent" to "dependent on morph" - and again, please use "morph", "morphotype" or "type" consistently throughout the manuscript.

Line 124 - "when group size increased" - is this from 1 to 2 to 6?

Line 126 - what is meant by "important"?

Line 128 - what is meant by "these"

Line 130-131 - "varied inter-morph"? Do you mean that this differed between morphs? And "inter-context" - does this mean that it differed between contexts?

Line 132 - again - what does "important" mean here?

Line 140 - "very different effects on the various sound categories". Does this mean the production of a different set of sounds? Please clarify.

Line 141 - what is "serial" and is it the same as "complex"? Please clarify.

Line 143 - "almost absent" - please clarify. See also line 158. What is a "level"?

Line 146 - variations in what?

Line 147 - should be "pose a hypothesis on the meaning...."

Line 151 - what does "over-represented" mean? Is this meant to be a quantitative statement? If so, are there statistics backing this up?

Line 154 - please provide more explanation for "resident-intruder assays", especially if they are as "appropriate" as stated.

Line 157 - was there only one dominant and one subordinate?

Line 160 - does "they" refer to Clicks? Please say this.

Line 163 - the sentence starting with "Of note...." Is very confusing. "Dominant fish" is mentioned at the end of the sentence - what is the behavioral context here?

Line 166 - what are "obstructions events"?

Line 168 - better to say - "aggressive behavior in SF was also tested using a classical mirror test..."

Line 170 - "ethogram" needs more explanation.

Line 173 - what is an "excellent correlation"? Please provide statistics.

Line 184 - Please change to "In contrast to _____, blind Pachon cavefish also producedHowever, 1) they...."

Line 192 - "perfusion of an odorant food solution" - please provide a bit more detail? Check this usage in figure captions. Concentration? Source of odorant?

Line 194 - "Single Clicks are shown as controls" - where? For purposes of comparison with other sounds?

Line 195 - What is "highly produced"?

Line 206 - "concomitant" - does this mean "simultaneously produced"

Line 209 - should be "must also have the same mechanical apparatus (sound production mechanism) to produce these sounds.."

Line 212 - "challenge the value?" - should this be "establish the meaning of Sharp Clicks...?"

Line 213 - should be "were exposed to sounds...."

Line 216 - what does "regrouped" mean?

Line 221 - what is meant by "important"?

Line 222 - should be "represent a transient abolition"

Line 231 - "their own" - this needs clarification. Should this be "sounds of the same morph"?

Line 239 - specify that it is "dark cave habitats".

Line 239 etc. - The discussion of the auditory system only cites one paper on adult blind cavefish [17] it is from 1970 - are there more recent data in the literature?

Line 240 - What is meant by "capacities" - how were they defined in the cited paper? The other cited paper [13] is on embryonic development (according to title), but it is likely post-embryonic development that might reveal changes in the cavefish.

Line 241 - what is meant by "maintenance of hearing" - no vertebrates have lost ears the way some have lost eyes; the idea that "soundscape" was initially the selection pressure for ear evolution and adaptation suggests that blind cavefish should have well-developed ears.

Line 243/4 - what is the evidence that these fish could not detect echolocation stimuli from bats? Astyanax is an otophysan and some in this larger taxon have high frequency hearing. The question should be - what is the morphology of cavefish vs. surface fish ears when compared to sighted outgroups (other characins - for which there IS data in the literature).

Line 246 - should be "now provides an answer to the puzzling question of what they are listening to:"

cavefish listen to their conspecifics.”

Table 1 - First line in left hand column indicates “ms” - milliseconds, but I believe the values indicated are expressed as seconds because they are all “...x10⁻³”. Best to check all units in statistical analyses.

Figures are very dense with information (due to space constraints of the journal?), but labelling of graphs and information rich caption content are critical.

Figure 1 - Audiograms and spectrograms are very small and very hard to see; the black backgrounds should be changed to white backgrounds if possible.

Figure 2 - good figure. Please check use of “solo”, “duo” - see comments about this above.

Figure 3 - “Evolution” - this is really a contrast between surface and cavefish. While this is an evolutionary transition via adaptation to a new environment, a discussion of evolution really needs an outgroup (a sighted, closely related characid?); “Evolution” per se should be elaborated upon in the discussion section

Supplemental Figure 2 - what are units on X-axes?

Supplemental Figure 3 - c and d need more explanation!

Supplemental Figure 4 - what are units on X-axes of small graphs? What is Delta?

Supplemental Audio recordings - are all of them necessary? Please provide rationale.

Reviewer #2:

Remarks to the Author:

This manuscript entitled “Evolution of acoustic communication in blind cavefish” by Hyacinthe et al., described extraordinary new insights for sound-based interactions within cave-dwelling or surface-dwelling forms of the Mexican tetra, *Astyanax mexicanus*. Those are: (1) both cave and surface morphs can elicit 6 major types of biological sounds, which include the ones that meet the criteria of fish vocalization (consisting of pulse series), (2) both morphs use these sound repertoires in different incidences, and under different ecological contexts, (3) a couple of sound types affect intra-morph’s behaviors (i.e. receivers changed their behavior according to the sound), and these behavioral responses (grouping together in surface morph; attraction toward the sound source in cavefish) were significantly relevant to their given ecological environments, and (4) One of these shifts of acoustic communications observed between surface and cave-dwelling forms is not simply based on the difference in visual sensitivity—indicating the evolution of acoustic communications.

Overall, this is a very well written manuscript with a clear logical-flow, and contains enough data evidence to support above conclusions. These conclusions will influence the wide fields including cave biology, evolutionary biology, animal behavior, animal communication, bioacoustics and sensory ecology. Also, given details of the materials and methods, other researchers are possible to reproduce the results.

I have no major critique but minors:

L41: “Chemosensory-triggered”

This is an ambiguous word. Please consider to mention that this chemical is from food such as “diet scent-triggered” or etc.

L45: “... and possibly speciation”

This is a little radical statement because these acoustic response does not show an obvious contribution for speciation (i.e. why the ‘grouping’ to ‘foraging’ shift could address animal speciation?). Please consider to exclude it.

L71: "hearing capabilities tested between 50 and 8000 Hz."

This is minor but please adjust as "50 and 7500 Hz" as written in Popper 1970.

L92: "(Fig. 1i, for Serial Clicks-like and Supplementary Audio Material 7-13 for Serial Clicks in the 6 caves; Fig.1j and Supplementary Audio Material 7-13 for the 6 sounds in the Pachón cave)."

Please revise the number of "supplementary audio materials" for Pachón cave. I believe it should be 8, 14-19 but authors need to recheck.

L132: "The most striking variations were an important use of Sharp Clicks in SF duos and SF groups after habituation (yellow on Fig. 2d)."

This yellow on Fig. 2d seems controversial with Fig 2f and g graph. Probably it is due to the y-axis's scales of Fig2f, g. If so, please note it in the main-text.

L136: "...this use has evolved in cavefish."

This conclusion is too early to be stated because the usage change of behavioral repertoires may be not observed after depriving visual stimulus in surface fish (authors wisely tested it in the following section, though). So, there is a chance that cavefish would had just expressed behavioral repertoire that surface fish (i.e. a proxy of surface-dwelling ancestor) naturally has in the dark, therefore, it cannot be called as evolution. Please revise.

L151: "Sharp Clicks were over-represented in SF duos and habituated SF groups"

Again, this statement was matched with Fig 2d but difficult to interpret from Fig 2f and 2g as I stated in L132.

L161: Supplementary Video1.

This file is difficult to see whether behavior and sound are coincident. Would it be possible to merge them into one video file?

L200: "These data suggest that, in cavefish, Sharp Clicks are chemosensory-triggered sounds produced during foraging, and they are especially used in conditions of low metabolic reserves."

Did you mean "These data suggest that, in cavefish, Sharp Clicks are foraging-related sounds, particularly used in starved condition"? The original sentence is slightly confusing.

L229: "This is consistent with the Sharp Clicks "feeding signal" triggering an attraction of CF towards the zone of emission of the sound."

Please reference Hüppop K (1987) Food-finding ability in cave fish. Int J Speleol 16:59-66.

Table1:

Please restate what Quartiles mean in the Table legend too. I saw its description in Materials and Methods but it is nicer to have it in the legend.

Figure2:

d: the significant stats symbols (a, b, c) are difficult to follow (within and between morphs, etc). please revise.

e-j are very complicated graphs. Please revise. Confusion seems to come from the inclusion of the grey-boxplot—SF duo new in the dark. Can you consider to have it in the far-right of each e-j panel?

This is one idea so author can decide to make it clearer.

Also, please remove ϕ because it is not significant.

Figure 3e is difficult to interpret. Was Position % of "mirror-far" compared with Sound occurrence % of

"sharp-clicks" or "rumblings"? How those are compared to calculate the stats? Please revise.

Figure 4c-n

Can you move one-second-bins panels to the supplemental data? All authors' conclusion can be seen in 30sec-bin panels and Fold-change panels.

Statistics

Not only interaction but please share all other stats values (F and P) for two-way ANOVA including morph and conditions. Also, please share stats scores for Mann-Whitney, Kruskal-Wallis and Friedman, Pearson correlation, PCA and permuted discriminant analysis. These data can be included in supplemental data.

Reviewer #3:

Remarks to the Author:

This study wants to compare the acoustic communication between blind and sighted morphs of the species *Astyanax mexicanus*, which is quite interesting. These fish were never recorded before although different trials have been realised. So, I was quite surprised the authors highlight 6 different kinds of sounds. They also determine the function for a single kind of sound (sharp click) and defend the hypothesis this sounds could have different meanings according to the fish way of life. They also try to provide the function of other sounds but it is not really convincing since different sounds would be produced during the same behavioral contexts, which is not usually the case. In most fishes, a type of sound corresponds to a behaviour or to different behaviours but, to the best of my knowledge, they are not different sounds related to a single behaviour. Moreover, the sound characterisation does not convince all these sounds are made by the fish. This is the reason I am afraid the authors could confound sounds that are used for acoustic communication and sounds that are by-products of fish activities such as swimming, manoeuvring, accelerating, eating, touching the wall, etc.

Communication between fish specimens is usually based on the production of trains of pulses or the repetition of single pulse but the authors seem to favour the use of single pulses to build most of the data (and the lack of temporal data does not allow to know the rate they are produced - What is the pulse period in train of pulses ? How many single pulses/time ? Etc.)...And this quite important for fishes living at night or in dark environment because the pulse repetition is used to certify the message. By experience many single pulses recorded in tanks and in the field are not done by the species. You can hardly be confident with the reception of a single pulse that is emitted from time to time. It results I have tried to analyse some sounds from supplementary data.

Suppl.2 can hardly be considered as a fish sound (and the author should apply the Akamatsu formula to delete frequencies higher than the resonant frequency of the tank). Moreover, I do not understand at all how it can be considered it is the same kind of sound that has been found in the field (Suppl. 15).

We would need more characterization to place suppl. 5, 7, 8, 9 and 10 in the same category. It lacks temporal data.

Suppl. 11 is quite interesting but really done by the fish, not an insect? It deserves deeper characterization.

Clock (suppl. 14 and suppl. 17)...How do you want to claim these sounds are the same kinds of sounds ? Also, the signal is really weak meaning it could be done by the fish or it could be background noise. I

understand the authors do their best to isolate the tank. Do they try to delete the electric noise coming from the lamps? If you want to convince, you have to clearly establish the conditions in which this kind of signal was done. Moreover, never forget a quiet tank simply does not exist, you always will have sounds you probably not hear but that the hydrophone is recording.

Supp 19. Where is the sound ? They are three sounds?

For all the sounds recorded in the field, how are you sure they are made by the fish and not by other living organisms ? Most of single pulse in the field can also be related to abiotic sounds or by by-product activities of living animals (eating, walking on the ground, escape, etc.).

Line 76. Why is the PCA done only on simple sounds ? Simple sounds are probably the worst to make this kind of analysis.

Line 90. « these 6 caves, and in a well hosting SF, sounds alike those identified in the lab were recorded 91(Fig.1i,for Serial Clicks-like and Supplementary Audio Material 7-13 for Serial Clicks in the 6caves; Fig.1jand Supplementary Audio Material 7-13for the 6 sounds in the Pachón cave) ». No statistical test ? No power spectrum for comparison ? From my experience, they certainly just cannot be the same because the tank walls modify the sounds...Always. The game is generally to find features allowing the link between sounds in tank and sounds from the field. Temporal data are the most reliable to do this.

Line 112. « In solo condition, Pachón emitted more sounds than SF for almost all sound ». This is the reason I have some doubts concerning the sounds. It is really unusual to have a single fish that makes sounds in solo condition. Here, it would make different kinds of sounds. Why is the fish making sounds ? Is the fish looking for partners ? If yes, a specific sound should correspond to this behaviour.

Lines 113 – 121. This paper wants to study the fish communication. What is the message corresponding to the different kinds of sounds ?

Line 135. « suggesting that the sound repertoire is used to convey information and that this use has evolved in cavefish ». Again, what kinds of information ?

Line 299 « The acoustic structure of sounds was characterized using a set of 1 temporal parameter, duration... » This way of doing is quite strange since it is usually admitted temporal features are the most important in fish acoustic communication. I am really surprised to not see information dealing with the pulse period. How can the author differ isolated pulses and series ? Are they series of pulses (that should be produced with regular periods) or can we find isolated pulses that are produced by groups. The message is completely different. The authors claim they have found six different sounds which is quite important for a given species. It means they have to show these sounds are clearly voluntary sounds used in fish communication.

The shape of the oscillogram should be the same for a given sound type. I do not see this. It is also true that the figure 1 is difficult to read. The different panels are too small, it is quite impossible to differentiate the different signals. In table 1, it is also strange to see frequencies >8000Hz. According to the size of the tank on the picture, I am quite sure the resonant frequency of the tank should be around 2500Hz, meaning authors should not take into account frequencies above 2500Hz.

In the ACP, It seems authors did not test the correlation between data to avoid overestimation? If features are correlated, they have to be removed.

Line 310. « Sounds used for these analyses were high pass filtered at 150 Hz. » So how is it possible

to have min freq that are below 150Hz ?

Table 1. duration, why 10^{-3} ? it would be 0.00X ms ?

Responses to Reviewers' comments for Hyacinthe et al. :

Reviewer #1:

The manuscript documents the exciting discovery of sound production in the surface and cave forms of the Mexican cave tetra, *Astyanax mexicanus*. This discovery further enriches our understanding of the sensory biology of this species, which has become a model species for genetics/genomics of development and sensory biology.

The work is described well, and the design of the work is very good. I cannot comment on the statistical analysis. Ability to reproduce work should be straightforward.

Thank you for this nice appreciation of our work.

Thank you also for all the suggestions that improve the manuscript. They are all addressed below:

Specific issues - pertaining to English word usage (for accuracy) and some scientific issues.

Line 25 - should be "allows the exchange of information..." **done**

Line 31 - chemosensory and mechanosensory should not be hyphenated **done**

Line 32 - "probably"?? - there is data in the literature. **Done**

Line 33 - citation 17 - this paper is from 1970 - is there more recent data? **No, this founding paper is the only one available about hearing capacities in *A. mexicanus*.**

Line 34 - how is "evolution" being defined in this paper? If the paper is only comparing surface vs. cavefishes, is "evolution" the best term? Is it a "shift"? What term was used in the papers about the lateral line-mediated VAB? See throughout summary and on line 58 as well. **This is an important point. The initial question of the study is indeed to study the "evolution" of acoustic communication, i.e., for example in the sense of a change in the repertoire, or as a compensation for the loss of visual communication in the dark. Therefore, we would like to keep that term in line 34 and some other places of the manuscript. However, the results show that there is indeed a "shift" in the use of sounds, which accompanies the behavioral changes observed in cavefish. Therefore, the terms "shift" or "change" have now been used in some other places.**

Line 36 - you didn't just "find" something - this is an important "discovery" - use that term. **Done**

Line 39 - is "morphotype" used to define the surface and cave "types", but please use the same term throughout the manuscript. **Done, we chose the use of "morph" throughout.**

Line 50 - perhaps say "...widely used among terrestrial and aquatic animals for..." **partly done**

Line 51 - should be “teleost fishes” - check throughout. **Done**

Line 53 - should be “behaviors are starting to be understood” **done**

Line 54 - should be “evolve via adaptation within species with respect to needs in their specific....”
Done

Line 56/57 - should be “which diverged.....experience markedly.....” **done**

Line 63 - say “First we determined...”. Then start next sentence with “Six different types....including 3 simple sounds and 3 complex sounds were recorded from 60 hours.....” **done**

Line 72 - the cited study used a behavioral assay - this should be mentioned. **Done**

Line 72/73 - This point needs to be made after the different sounds are defined...in the results. **Done**

Line 75 - what is meant by “to validate human hear identification”? **this was un-necessary and has been removed.**

Line 84 - Is “or” correct? **Removed**

Line 86 - should be “caves with....”; also in line 91 - should be “in a well in which surface forms are found”. **Done**

Line 94 - What “repertoire” is being referred to here? **It is the repertoire of six sounds, now indicated.**

Line 96 - should be “pursued additional analysis of sounds in.....” **done**

Line 101 - elaborate on the implications of absence of a visual modality, e.g., “causing them to be more reliant on other sensory systems”. **Done.**

Line 101-102 - Evolution of acoustic communication in cavefish??? Or are the differences between surface fish and cavefish the focus? This is confusing. **In response to comment above, “evolution” has been replaced by “changes”. We do not understand this comment of the reviewer. The goal of the paper is to infer changes which have occurred in the cavefish lineage, by a comparative analysis with extant surface fish (in a sense, and although improperly, “assimilated” to the ancestral state).**

Line 106 - should be “used for acoustic communication..., then sound production should vary....”
Done

Line 108 - instead of “solo”, “duo” which is awkward, please use “individual fish”, “pairs of fish”, “groups of 6 fish”. Check this throughout. We have now given the definition of solo/duo/group according to the reviewer’s suggestion, but we would like to keep the short terms for the following text for the sake of simplicity of writing, and the small size in the figures.

Line 108 - I think that “discovery” should be “exploration”, which is a term used in the literature.
done

Line 121 - change “morph-dependent” to “dependent on morph” - and again, please use “morph”, “morphotype” or “type” consistently throughout the manuscript. Done, see also response above.

Line 124 - “when group size increased” - is this from 1 to 2 to 6? Yes, this is now indicated.

Line 126 - what is meant by “important”? has been changed to “substantial”

Line 128 - what is meant by “these” it is now indicated: in solo/duo/group conditions

Line 130-131 - “varied inter-morph”? Do you mean that this differed between morphs? And “inter-context” - does this mean that it differed between contexts? Yes, this is what we meant. We have reformulated to avoid confusion.

Line 132 - again - what does “important” mean here? Has been changed to “significant”

Line 140 - “very different effects on the various sound categories”. Does this mean the production of a different set of sounds? Please clarify. Has been changed to “Absence of light had very different effects on the 6 sound types”

Line 141 - what is “serial” and is it the same as “complex”? Please clarify. As defined in the first paragraph of results (and now further documented and analyzed in the present revised version in response to another reviewer), we have divided sounds in two categories: “simple” sounds, which correspond to single pulses of short duration (<20msec), and “complex” sounds, which correspond to multipulses (with pulses being separated by intervals inferior to 1sec) or long pulses > 20msec. According to this, and as shown in Fig. 1, a “serial click” is a complex sound that is composed of a repetition of clicks separated by intervals of less than 1sec.

Line 143 - “almost absent” - please clarify. See also line 158. What is a “level”? Sorry for these approximations, changed to “very strongly reduced in the dark” and “reaching the scores of Pachón CF duos”

Line 146 - variations in what? It is now indicated: “the variations in the production of sounds...”

Line 147 - should be “pose a hypothesis on the meaning.....” **done**

Line 151 - what does “over-represented” mean? Is this mean to be a quantitative statement? If so, are there statistics backing this up? **This sentence is a summary of Fig.2d and 2g, on which statistics (Fisher tests and 2-way ANOVA tests, respectively) have been done, both in terms of the use of the Sharp Clicks with respect to other sounds, and in terms of the variations in the numbers of Sharp Clicks produced in the solo/duo/group/dark condition. We have used the word “over-represented” as a way to summarize all these data, as a link to start the next paragraph.**

Line 154 - please provide more explanation for “resident-intruder assays”, especially if they are as “appropriate” as stated. **We used the word “appropriate” to express the fact that the resident intruder test, as opposed to the mirror test that we have also used in our study, does not necessitate vision. This is now explained in text: “To test this hypothesis, we performed resident-intruder assays, an appropriate way to compare aggressiveness in eyed and eyeless animals because it does not require vision [9, 26]. In brief, individual fish were habituated in a tank overnight, and the next day one fish (the intruder) was transferred in the tank of the other (the resident).”**

Line 157 - was there only one dominant and one subordinate? **Yes, there are 2 fish in the test tank, one will progressively become the dominant and the other the subordinate during the one-hour test. This is also explained in Methods.**

Line 160 - does “they” refer to Clicks? Please say this. **Yes, done.**

Line 163 - the sentence starting with “Of note....” Is very confusing. “Dominant fish” is mentioned at the end of the sentence - what is the behavioral context here? **The sentence has been cut and rephrased to remove any potential confusion. The behavioral context is the resident-intruder assay, like for this whole paragraph.**

Line 166 - what are “obstructions events”? **An obstruction event was defined as a fish blocking or obstructing the other in a corner or restricted part of the aquarium, or preventing it from escaping. This explanation was lacking and has now been added in Methods.**

Line 168 - better to say - “aggressive behavior in SF was aslo tested using a classical mirror test...” **done**

Line 170 - “ethogram” needs more explanation. **Ethogram is now defined at the first occurrence of this term, i.e., line 109 (“graphs depicting the production of sounds or behaviors along time”)**

Line 173 - what is an “excellent correlation”? Please provide statistics. **All statistics related to Fig.3f are now given in Suppl. Table 2 (also in response to reviewer 2). The Pearson’s coefficient for the Sharp Click/Attacks correlation is also now indicated in text: 0.64.**

Line 184 - Please change to "In contrast to _____, blind Pachon cavefish also produced ...However, 1) they.....". We do not agree with this particular suggestion "In contrast to", which would affect the meaning of the text. The second proposed modification (=However) has been done.

Line 192 - "perfusion of an odorant food solution" - please provide a bit more detail? Check this usage in figure captions. Concentration? Source of odorant? Done. The concentration is indicated in Methods. "crushed and filtered granular fish food" has been added in text.

Line 194 - "Single Clocs are shown as controls" - where? For purposes of comparison with other sounds? Single Clocs are shown together with Sharp Clicks on Fig. 3h, in order to compare the variations in the sound we are interested in (the Sharp Clicks, which vary according to chemosensory stimulation and foraging behavior) with a "control sound" (the Clocs) which do not vary in these same conditions of stimulation and behavioral expression.

Line 195 - What is "highly produced"? Changed for strongly produced

Line 206 - "concomitant" - does this mean "simultaneously produced" Yes, we have changed the wording.

Line 209 - should be "must also have the same mechanical apparatus (sound production mechanism) to produce these sounds... We don't understand the change asked by the reviewer, we would rather keep the initial writing.

Line 212 - "challenge the value?" - should this be "establish the meaning of Sharp Clicks..."? OK, change done.

Line 213 - should be "were exposed to sounds..." Done

Line 216 - what does "regrouped" mean? Sorry. Changed to gathered.

Line 221 - what is meant by "important"? Changed to significant.

Line 222 - should be "represent a transient abolition" done

Line 231 - "their own" - this needs clarification. Should this be "sounds of the same morph"? The statement was meant to be more general, i.e., that *A. mexicanus* are responsive in general to the types of sounds they produce (i.e., the common repertoire of 6 sounds cavefish and surface fish have). However, we have not tested the response of cavefish to surface fish sounds and vice versa, nor have we tested the sound production in a tank containing two fish, one surface and one cavefish.

The sentence has therefore been changed for clarity: “these data indicate that both SF and CF show behavioral responses to sounds produced by conspecifics of their own morphotype”

Line 239 - specify that it is “dark cave habitats”. Done.

Line 239 etc. - The discussion of the auditory system only cites one paper on adult blind cavefish [17] it is from 1970 - are there more recent data in the literature? No, as already stated in response to the same question on line 33, to our knowledge there is no other data on hearing abilities in *Astyanax*.

Line 240 - What is meant by “capacities” - how were they defined in the cited paper? The other cited paper [13] is on embryonic development (according to title), but it is likely post-embryonic development that might reveal changes in the cavefish. Actually, “capacities” is the term used by Popper himself in the cited 1970 paper. It refers both to the sensitivity or threshold (decibels), and to the bandwidth (frequencies) that the fish are able to detect. The audiograms of the two *Astyanax* morphs were not significantly different, if not identical (Fig. 2 in Popper 1970). This has been added in text. The other paper cited is indeed about embryonic development of the otic placode. No difference in size were observed between the two morphs (although major differences were observed for other placodes, the lens and olfactory placodes, at the same stages). Two references about adult comparative morphology of the Weberian ossicles (Popper 1971) and the labyrinth (Schemmel 1967) have also been added. The absence of anatomical difference at embryonic and adult stages, together with the absence of functional differences in adults, strongly suggests that the auditory system did not undergo major changes during cavefish evolution in the dark (again, as opposed to other sensory systems).

Line 241 - what is meant by “maintenance of hearing” - no vertebrates have lost ears the way some have lost eyes; the idea that “soundscape” was initially the selection pressure for ear evolution and adaptation suggests that blind cavefish should have well-developed ears. Popper (1970), followed by Soares et al. (2016), discussed the maintenance of significant auditory capacities in cavefish: “The significance of the extensive auditory capacity of the cave animal for survival in its natural environment is unclear”. Popper proposed that it could be maintained because of pleiotropy (“the interrelationships between the ear and other structures that start to develop early in the fish are so complex that a gene altering the ear would disastrously alter the other organ systems”), or else because the ears may help the fish maintain equilibrium and compensate for eye loss, as suggested in cave *Amblyopsids*. However, this was before our discovery that *A. mexicanus* is a sonic species. This is why we discussed this idea the way we did.

Line 243/4 - what is the evidence that these fish could not detect echolocation stimuli from bats? *Astyanax* is an otophysan and some in this larger taxon have high frequency hearing. The question should be - what is the morphology of cavefish vs. surface fish ears when compared to sighted outgroups (other characins - for which there IS data in the literature). Regarding high frequency hearing, the well-studied case among teleosts –and the only we are aware of in the literature- is for clupeids, which can show sensitivity up to 180kHz. This unique property is thought to be possible thanks a specialized structure shared by this family of teleosts, the « utricle », in their inner ear. However, clupeids are not otophysans. Although otophysans are also supposed to be « hearing specialists » thanks to their Weberian apparatus (connecting the swim bladder, the sound detector,

to the inner ear), their range of sensitivity goes up to 10kHz or so, but we have not found evidence of higher frequency hearing for this super-order. The sentence in the manuscript has been changed to be even more speculative, and a reference on high frequency hearing in clupeids has been added (Popper, 2004).

For the second part of the comment, unfortunately, there is no detailed comparative study on ear morphology on the two *Astyanax* morphotypes. Popper (1971) compared the morphology of the Weberian ossicles (connecting the swim bladder, the sound detector, to the inner ear) in the two morphs and he discovered some inter-morph as well as some intra-morph differences. He concluded that *"the significance of the variations, if any, was not evident"*, especially with regards to his own previous hearing capacity study (Popper, 1970). In a remark in this paper, he also stated that the *"swim bladder and inner ear showed no inter-specific differences"*. Schemmel (1967) also briefly reported *"no degenerative tendencies or other deviations in the labyrinth or in the Weberian apparatus of the cavernicolous forms compared to the epigeal fish"*. But these fragmentary data are difficult to compare to other characins, especially in the context of a discussion on the likelihood/hypothesis of high frequency hearing.

Line 246 - should be "now provides an answer to the puzzling question of what they are listening to: cavefish listen to their conspecifics." **done**

Table 1 - First line in left hand column indicates "ms" - milliseconds, but I believe the values indicated are expressed as seconds because they are all "...x10⁻³". Best to check all units in statistical analyses. **Corrected.**

Figures are very dense with information (due to space constraints of the journal?), but labelling of graphs and information rich caption content are critical.

Figure 1 - Audiograms and spectrograms are very small and very hard to see; the black backgrounds should be changed to white backgrounds if possible. **The figure 1 has been split into 2 figures, to increase the size of the sonograms and graphs. Moreover, magnifications are provided as insets, to appreciate details of the sonogram for all sounds that do not spread to frequencies higher than 5kHz (i.e., only Clicks are not magnified). However, we would prefer keeping the black background as we prefer its contrast and also "art" aspect.**

Figure 2 - good figure. Please check use of "solo", "duo" - see comments about this above. **Done, see above.**

Figure 3 - "Evolution" - this is really a contrast between surface and cavefish. While this is an evolutionary transition via adaptation to a new environment, a discussion of evolution really needs an outgroup (a sighted, closely related characid?); "Evolution" per se should be elaborated upon in the discussion section. **OK. The title of the figure has been changed to : "Comparison of Sharp Clicks use and significance in cavefish and surface fish".**

Supplemental Figure 2 - what are units on X-axes? **Y-axis is the number of sounds produced and X-axis is the number of agonistic events observed (now Suppl. Fig 3).**

Supplemental Figure 3 - c and d need more explanation! **Done**

Supplemental Figure 4 - what are units on X-axes of small graphs? What is Delta? The legend has now been corrected with more explanations, and the X axis has been added on the figure (time).

Supplemental Audio recordings - are all of them necessary? Please provide rationale. Yes, we think all of them are necessary, for 3 reasons. (1) to fully convince the readers (and the reviewers) that *Astyanax mexicanus* is a sonic species, both in the wild and in the lab, and has the same repertoire of sounds in the natural and laboratory conditions. (2) to illustrate broadly the sampling of sounds we have been able to perform in the field. (3) to allow the reader to appreciate the variety of sounds produced, for example from a few to more than 80 Click pulses in a Serial Clicks, "in real" with the headphones or from a speaker, instead of "merely" on a sonogram.

Reviewer #2

This manuscript entitled “Evolution of acoustic communication in blind cavefish” by Hyacinthe et al., described extraordinary new insights for sound-based interactions within cave-dwelling or surface-dwelling forms of the Mexican tetra, *Astyanax mexicanus*. Those are: (1) both cave and surface morphs can elicit 6 major types of biological sounds, which include the ones that meet the criteria of fish vocalization (consisting of pulse series), (2) both morphs use these sound repertoires in different incidences, and under different ecological contexts, (3) a couple of sound types affect intra-morph’s behaviors (i.e. receivers changed their behavior according to the sound), and these behavioral responses (grouping together in surface morph; attraction toward the sound source in cavefish) were significantly relevant to their given ecological environments, and (4) One of these shifts of acoustic communications observed between surface and cave-dwelling forms is not simply based on the difference in visual sensitivity—indicating the evolution of acoustic communications. Overall, this is a very well written manuscript with a clear logical-flow, and contains enough data evidence to support above conclusions. These conclusions will influence the wide fields including cave biology, evolutionary biology, animal behavior, animal communication, bioacoustics and sensory ecology. Also, given details of the materials and methods, other researchers are possible to reproduce the results.

Thank you for this enthusiastic appreciation of our work.

Thank you also for all the suggestions that improve the manuscript, they are all addressed below:

I have no major critique but minors:

L41: “Chemosensory-triggered”

This is an ambiguous word. Please consider to mention that this chemical is from food such as “diet scent-triggered” or etc. **Done**

L45: “... and possibly speciation”

This is a little radical statement because these acoustic response does not show an obvious contribution for speciation (i.e. why the ‘grouping’ to ‘foraging’ shift could address animal speciation?). Please consider to exclude it. **Agreed. The statement has been removed from the abstract, and is now moved to the discussion (with a short explanation in a new paragraph).**

L71: “hearing capabilities tested between 50 and 8000 Hz.”

This is minor but please adjust as “50 and 7500 Hz” as written in Popper 1970. **Done**

L92: “(Fig. 1i, for Serial Clicks-like and Supplementary Audio Material 7-13 for Serial Clicks in the 6 caves; Fig.1j and Supplementary Audio Material 7-13 for the 6 sounds in the Pachón cave).”

Please revise the number of “supplementary audio materials” for Pachón cave. I believe it should be 8, 14-19 but authors need to recheck. **Yes, done**

L132: “The most striking variations were an important use of Sharp Clicks in SF duos and SF groups after habituation (yellow on Fig. 2d).”

This yellow on Fig. 2d seems controversial with Fig 2f and g graph. Probably it is due to the y-axis’s

scales of Fig2f, g. If so, please note it in the main-text. OK, in fact the two representations are different and the results are not controversial: Fig.2d shows the usage of the different sounds in different conditions, i.e., the yellow bar shows the proportion of all the sounds produced that corresponds to Sharp Clicks. Fig.2g on the other hand shows the absolute total numbers of sounds produced during a given period of time. Thus, even though SF groups after habituation do not produce a lot of sounds in total (Fig.2c), almost half of these sounds are Sharp Clicks, therefore the yellow bar is large in Fig.2d.

L136: "...this use has evolved in cavefish."

This conclusion is too early to be stated because the usage change of behavioral repertoires may be not observed after depriving visual stimulus in surface fish (authors wisely tested it in the following section, though). So, there is a chance that cavefish would had just expressed behavioral repertoire that surface fish (i.e. a proxy of surface-dwelling ancestor) naturally has in the dark, therefore, it cannot be called as evolution. Please revise. Done, "has evolved" has been changed to "is changed".

L151: "Sharp Clicks were over-represented in SF duos and habituated SF groups"

Again, this statement was matched with Fig 2d but difficult to interpret from Fig 2f and 2g as I stated in L132. Please see response given above, for the remark on line 132.

L161: Supplementary Video1.

This file is difficult to see whether behavior and sound are coincident. Would it be possible to merge them into one video file?

Sorry, but this Suppl is already a single file combining the video and sound track (mp4). This compressed format is necessary for the submission site.

L200: "These data suggest that, in cavefish, Sharp Clicks are chemosensory-triggered sounds produced during foraging, and they are especially used in conditions of low metabolic reserves."

Did you mean "These data suggest that, in cavefish, Sharp Clicks are foraging-related sounds, particularly used in starved condition"? The original sentence is slightly confusing. Corrected.

L229: "This is consistent with the Sharp Clicks "feeding signal" triggering an attraction of CF towards the zone of emission of the sound."

Please reference Hüppop K (1987) Food-finding ability in cave fish. Int J Speleol 16:59–66. Done. Thank you very much for this insightful suggestion.

Table1:

Please restate what Quartiles mean in the Table legend too. I saw its description in Materials and Methods but it is nicer to have it in the legend. Done

Figure2 (now Fig3):

d: the significant stats symbols (a, b, c) are difficult to follow (within and between morphs, etc). please revise. Done, we have modified the graph, so that the significant differences are easier to catch.

e-j are very complicated graphs. Please revise. Confusion seems to come from the inclusion of the

grey-boxplot—SF duo new in the dark. Can you consider to have it in the far-right of each e-j panel? This is one idea so author can decide to make it clearer. We have thought about moving the grey bars to the far right, but this posed other problems for interpretation of the figure and for the inclusion of the statistics. Thus, to improve we decided to add a grey shade on the entire “dark SF” condition in duo, and to add ticks on the horizontal x axis in order to better separate the different conditions. We hope this looks better now.

Also, please remove ϕ because it is not significant. Done.

Figure 3e (now Fig 4f) is difficult to interpret. Was Position % of “mirror-far” compared with Sound occurrence % of “sharp-clicks” or “rumblings”? How those are compared to calculate the stats? Please revise. Fig.4f shows transition analysis, while Fig.4g shows correlation matrix. In the transition analysis, the position (mirror far or mirror close) and the sounds produced (Clocs, Sharp Clicks or Rumblings) in the second just before or just after an attack are plotted. For example, in ~30% of the cases, attacks were just preceded or followed by the emission of a Sharp Click (yellow bar). Note that all statistical analyses are now provided in Supplemental Table 6, as also requested below in the last point.

Figure 4c-n (now Fig 5)

Can you move one-second-bins panels to the supplemental data? All authors’ conclusion can be seen in 30sec-bin panels and Fold-change panels. We would rather keep it in the main Figure (but note that all the sound production data have been moved to Suppl.). Actually, we think the one second bin time-course representations (left panels for each morph) give a good view of the strong reproducibility of the behaviors and of the immediate effects of sound stimulations on the fish behaviors.

Statistics

Not only interaction but please share all other stats values (F and P) for two-way ANOVA including morph and conditions. Also, please share stats scores for Mann-Whitney, Kruskal-Wallis and Friedman, Pearson correlation, PCA and permutated discriminant analysis. These data can be included in supplemental data. All statistical analyses are now provided in new Supplemental Table 6, as now indicated in the “Statistics” section of the Methods.

Reviewer #3 :

This study wants to compare the acoustic communication between blind and sighted morphs of the species *Astyanax mexicanus*, which is quite interesting. These fish were never recorded before although different trials have been realised. So, I was quite surprised the authors highlight 6 different kinds of sounds. They also determine the function for a single kind of sound (sharp click) and defend the hypothesis this sounds could have different meanings according to the fish way of life. They also try to provide the function of other sounds but it is not really convincing since different sounds would be produced during the same behavioral contexts, which is not usually the case. In most fishes, a type of sound corresponds to a behaviour or to different behaviours but, to the best of my knowledge, they are not different sounds related to a single behaviour.

> We disagree with this statement: a given context (here, solo, or duo, or group) encompasses different behaviors, hence it is expected that different sounds can be produced. For example, in a group agonistic behaviors, establishment of hierarchy, schooling, kin/non-kin recognition, sleeping, exploration and locomotion, or else reproductive behaviors, can all arise.

Moreover, the sound characterisation does not convince all these sounds are made by the fish. This is the reason I am afraid the authors could confound sounds that are used for acoustic communication and sounds that are by-products of fish activities such as swimming, manoeuvring, accelerating, eating, touching the wall, etc.

> Our experiments and analyses were controlled to avoid this type of mistakes. First, concerning laboratory recordings, it was quite easy to sort the sounds from the "background noises" as we always had simultaneous audio and video recordings. Sounds due to touching the walls (or the hydrophone) were recognized and excluded. Of note, there are not so many shocks on the walls, including in eyeless cavefish, due to their enhanced hydrodynamic imaging abilities (Windsor et al. 2008). Moreover, no food was provided, so the sounds could not come from this activity (even when food odor was delivered, in Figure 4, it was crushed and filtered so could in theory not elicit sounds). Second, concerning field recordings, we have recorded in different natural pools, either in a net or in freely swimming fish (Suppl. Figure), and the two conditions yielded the same sounds, similar to those recorded in the lab. Moreover, we also have audio and video recordings of wild cavefish in plastic pools (1m x 1m x 0.2m of cave water), for experimentations on chemical reception (similar to Bibliowicz et al., *Evo Devo* 2013). The sounds heard in the plastic pools were similar to those recorded in the lab. Lastly, both in the plastic pools and in the lab condition, there was of course no other fish species, insects or invertebrates, and the sounds recorded can only be from the fish.

Communication between fish specimens is usually based on the production of trains of pulses or the repetition of single pulse but the authors seem to favour the use of single pulses to build most of the data (and the lack of temporal data does not allow to know the rate they are produced - What is the pulse period in train of pulses ? How many single pulses/time ? Etc.)...And this quite important for fishes living at night or in dark environment because the pulse repetition is used to certify the message. By experience many single pulses recorded in tanks and in the field are not done by the species. You can hardly be confident with the reception of a single pulse that is emitted from time to time. It results I have tried to analyse some sounds from supplementary data.

> We would like to stress that the entire Figure 3 (previous Figure 2) precisely corresponds to temporal analysis: this figure shows the number of sounds produced by units of time, in different conditions. For example, the fish produce about 10 single clicks per hour, and about 5 serial clicks per hour.

As now clearly indicated in Methods and Results, pulses were considered “simple” or “single” if they were of short duration (<20msec) and separated by >1sec interval from the next pulse (threshold defined from the histogram of the inter-pulse durations). Further, to answer more precisely the reviewer’s question on pulse rate, we have re-analyzed serial clicks and serial clics. This pulse rate analysis was performed on a total of n=186 sounds, coming from 8-11 different fish (for each morphotype and each sound type; See new Suppl. Table 2) and included into a PCA. These additional data are now inserted in the Results (Fig.1i). Briefly, they show that *“Serial Clicks and Serial Clics belonged to separate clusters on the grounds of pulse rate parameters, and for both morphs (Fig. 1i and Supplementary Table 2). These results support the hypothesis that the different simple sounds and complex sounds identified could carry different information and could be used differently according to context and behavior”*.

The statement that single pulses are usually not produced by the species is a bit upsetting: as already indicated above, video controls have always been used, and there was obviously nothing else than *A. mexicanus* fish in the recording tanks. Moreover, our deep behavioral analysis focused on sharp clicks (i.e., a simple sound) clearly demonstrates the relevance of single pulses with respect to behaviors.

Finally, in most fish acoustic studies, authors have concentrated their efforts on the most complex signals, which are in general trains of pulses. Here we decided to pay attention to all the sounds produced by the fish. We were inspired by recent papers on birds, showing the importance of the calls, i.e., sounds shorter and simpler than songs, in acoustic communication (see for instance, Fernandez MSA, Vignal C, Soula HA, 2017. Impact of group size and social composition on group vocal activity and acoustic network in a social songbird. *Animal Behaviour* 127 : 163-178).

Suppl.2 can hardly be considered as a fish sound (and the author should apply the Akamatsu formula to delete frequencies higher than the resonant frequency of the tank). Moreover, I do not understand at all how it can be considered it is the same kind of sound that has been found in the field (Supp. 15).

> Suppl. 2 is a single Click recorded in the lab. Suppl. 15 is a single Click recorded in the wild, from the Pachón cave.

The Akamatsu formula has not been applied on Suppl. 2, please see the detailed explanations for why we did not, in the response to a question about Table 1 below.

The question of the similarity between lab-recorded and wild-recorded sounds is important. Regarding specifically the Clicks, it is true that lab sounds span the whole range of frequencies up to 20kHz, whereas wild sounds span a shorter range of frequencies, up to 10kHz.

First, we would like to argue, as the reviewer must know, that recording conditions in the field are different (not the same volume of water around the emitting fish, therefore different refraction) and they have not been obtained with the same equipment. The wild recordings show a higher background noise, please compare the traces in the upper panel of each sonogram in Fig.1 and Fig.2. However, it is clear that all the wild Clicks (here, serial Clicks) recorded from the 7 different locations (including the Pachón cave and the well with surface fish) are more similar to each other than to lab-

recorded Pachón or surface fish Clicks and serial Clicks. This strongly supports the idea that the apparent difference in the Clicks from the wild and from the lab is due to the recording conditions.

Second, to answer more precisely the reviewer's question on the similarity between lab- and wild-recorded sounds, we have now extracted the physical parameters from 45 single Clicks and 44 single Clocs produced by wild Pachón cavefish (Suppl. Table 1), and we have performed a PCA comparing these wild sounds together with the simple sounds recorded in the lab. These new data are now included in Figure 2c. Briefly, they show that *"1) in the wild also, Single Clicks and Single Clocs were easily discernable and corresponded to distinct sounds, and 2) sounds produced in the wild were alike those produced in laboratory conditions, and grouped in clusters in the ACP."*

Third, we have strong unpublished evidence that will be released in another manuscript on sound production mechanisms, that the Clicks (single and serial) are sounds that are produced voluntarily by the fish, after capturing an air bubble at the surface of the water. This explains why these sounds may "sound like noise" and why the reviewer is skeptical that they are produced by the fish. This also explains the difficulty in comparing the Click pulses: the air bubbles captured are probably not always the same in terms of size, and their use in the production of sounds may also vary, in time and in the amount of air released at a time. The variability of lab Clicks is indeed visible in the PCA in Figure 1 and in the Table 1. Interestingly Nelson (1964) reported this way of producing sound in a small characid (*Glandulocauda inequalis*) and showed that it was part of the male courtship behavior. He reported that, probably due to the mode of production of the sound, the different pulses could *"vary considerably in frequency even within one "Croak"*. He also showed that this had nothing to do with oxygenation or breathing problems, and hypothesized that this behavior may derive from a behavior where the fish eats food at the surface of the water.

Finally, we certainly do not exclude the possibility that there could be different sub-categories of Clicks, or individual signatures, or cave-specific signatures, but this is clearly out of scope of this paper and will be investigated in the future.

We would need more characterization to place suppl. 5, 7, 8, 9 and 10 in the same category. It lacks temporal data.

> Again, about temporal data, please see responses above and novel analyses performed to answer the point.

Suppl. 11 is quite interesting but really done by the fish, not an insect? It deserves deeper characterization.

> We have never observed crayfish or other insects in the Tinaja main pool where the recordings have been performed. Moreover, and again, there was nothing else than cavefish in the plastic pools used for some wild recordings, and the similarity between wild sounds and lab sounds (where there is obviously no insect in the tank) supports that the sounds can only be produced by the fish.

Clock (suppl. 14 and suppl. 17)...How do you want to claim these sounds are the same kinds of sounds? Also, the signal is really weak meaning it could be done by the fish or it could be background noise. I understand the authors do their best to isolate the tank. Do they try to delete the electric noise coming from the lamps? If you want to convince, you have to clearly establish the conditions in which this kind of signal was done. Moreover, never forget a quiet tank simply does not exist, you always will have sounds you probably not hear but that the hydrophone is recording.

> Suppl 14 and Suppl 17 are wild-recorded sounds (single Cloc and serial Cloc, respectively) in the Pachón cave. Thus, it was not obtained from a tank, and there were no lamps around. Please see the argumentation above for the same question on Suppl. 2 / Clicks.

Regarding the technical aspects of lab recordings, yes we have done our best: the hydrophone and speaker wires were as far as possible from the 220V wires, and as far as possible from each other. There was no light/bulb on the aquarium, the light was from the room. Finally, on the audio bands from background recordings (no fish in the tank), the background never shows pulsed sounds in a short frequency range like the Clocs.

Please, see below on the figure specially prepared to convince the reviewer, that the background noise recorded with no fish in the tank cannot be mixed up with fish sounds, especially after having applied a high pass filter at 150Hz:

Supp 19. Where is the sound ? They are three sounds?

> Suppl. Audio 19 (Pachón wild Rumbling) has been corrected and replaced. Please note that it is better to listen to all these Suppl Audio samples with headphones, rather than with the computer's speakers.

For all the sounds recorded in the field, how are you sure they are made by the fish and not by other living organisms ? Most of single pulse in the field can also be related to abiotic sounds or by by-product activities of living animals (eating, walking on the ground, escape, etc.).

> Again, this point has already been addressed and answered above.

Line 76. Why is the PCA done only on simple sounds ? Simple sounds are probably the worst to make this kind of analysis.

> This point has also already been addressed above. We have concentrated our efforts, both for sound analyses and for behavioral function, on simple sounds in this paper.

Line 90. « these 6 caves, and in a well hosting SF, sounds alike those identified in the lab were recorded 91(Fig.1i,for Serial Clicks-like and Supplementary Audio Material 7-13 for Serial Clicks in the 6caves; Fig.1jand Supplementary Audio Material 7-13for the 6 sounds in the Pachón cave) ». No statistical test ? No power spectrum for comparison ? From my experience, they certainly just cannot be the same because the tank walls modify the sounds...Always. The game is generally to find features allowing the link between sounds in tank and sounds from the field. Temporal data are the most reliable to do this.

> Again, and as already explained above, to answer more precisely the reviewer's question on the similarity between lab- and wild-recorded sounds, from 44 single Clicks and 45 single Clocs produced by wild Pachón cavefish (Suppl. Table 1), and we have performed a PCA comparing these wild sounds together with the simple sounds recorded in the lab. These new data are now included in Figure 2c. Briefly, they show that "1) in the wild also, Single Clicks and Single Clocs were easily discernable and corresponded to distinct sounds, and 2) sounds produced in the wild were indistinguishable from those produced in laboratory conditions."

Line 112. « In solo condition, Pachón emitted more sounds than SF for almost all sound ». This is the reason I have some doubts concerning the sounds. It is really unusual to have a single fish that makes sounds in solo condition. Here, it would make different kinds of sounds. Why is the fish making sounds ? Is the fish looking for partners ? If yes, a specific sound should correspond to this behaviour.

> It is true that we do not know yet the "meaning" of each sound we have identified, as we have concentrated on Sharp Clicks and this lead us to deep behavioral investigations and a lot of produced data.

Our study is precisely based on the idea that, if the sound has a behavioral meaning or function, its use should vary depending on the context. And the idea that a given condition can encompass different behaviors has already been discussed above. In the case of the solo context, exploration/locomotion, stress, sleeping, seeking for partners, or other behaviors we cannot think about, can all arise. The "looking for partners" hypothesis is indeed plausible and will be worth testing in the future.

Lines 113 – 121. This paper wants to study the fish communication. What is the message corresponding to the different kinds of sounds ?

> Again, we do not know yet the "meaning" of each sound we have identified, as we have concentrated on Sharp Clicks and this lead us to deep behavioral investigations and a lot of produced data. Please, consider that the amount of data, characterization and novelty present in the paper is already very comprehensive.

Line 135. « suggesting that the sound repertoire is used to convey information and that this use has evolved in cavefish ». Again, what kinds of information ?

> This sentence is used as a "linker" between two paragraphs in the manuscript, with the term "suggesting", to state the hypotheses that will be tested in the next paragraphs.

Again, we believe that we bring compelling evidence that “*Sharp Clicks are visually-triggered sounds produced by dominant surface fish during agonistic behavior and chemosensory, food odor-triggered sounds produced by cavefish during foraging behavior, and which also elicits different reactions in the two morphs in play-back experiments*”. Thus, the sound is indeed used to convey information and its use has evolved in cavefish along with the evolution of behaviors.

Line 299 « The acoustic structure of sounds was characterized using a set of 1 temporal parameter, duration... » This way of doing is quite strange since it is usually admitted temporal features are the most important in fish acoustic communication. I am really surprised to not see information dealing with the pulse period. How can the author differ isolated pulses and series ? Are they series of pulses (that should be produced with regular periods) or can we find isolated pulses that are produced by groups. The message is completely different.

> This point again refers to temporal analysis and pulse rate analysis, please see the responses that we have already given above.

The authors claim they have found six different sounds which is quite important for a given species. It means they have to show these sounds are clearly voluntary sounds used in fish communication.

> We have indeed identified 6 sounds and we seek at understanding whether they are all used for communication purposes. We demonstrated convincingly that at least one of them, the Sharp Click, is used for acoustic communication. But nowhere did we state that it is the case for the 5 other sound types. For example, the Clicks and serial Clicks do not vary according to the different contexts we have tested (Fig. 3), which is contradictory to the rationale presented in the first sentences of the paragraph “*Sound production and social interactions*”: *We predicted that, if sounds produced by A. mexicanus are used for acoustic communication, then production should vary according to the social context*. Thus, the significance of the 5 other sounds remains an open question and is the focus of our next investigations.

The shape of the oscillogram should be the same for a given sound type. I do not see this. It is also true that the figure 1 is difficult to read. The different panels are too small, it is quite impossible to differentiate the different signals.

>OK. To improve this, the original Figure 1 has been split in two Figures, which allowed enlarging the panels. Moreover, magnifications of pulses in sonagrams are now given in insets, which allows better comparing sounds, and appreciating their differences and specificities, in addition to Table 1 (and Suppl Tables 1 and 2) giving their detailed physical parameters and the PCAs showing their proper categorization.

In table 1, it is also strange to see frequencies >8000Hz. According to the size of the tank on the picture, I am quite sure the resonant frequency of the tank should be around 2500Hz, meaning authors should not take into account frequencies above 2500Hz.

> Please consider the 2 points below:

1.- We have put some foam on the walls (except on the front one, for video recording) to attenuate sound reflections and to limit the excitation of the frequency resonance of the tank.

2.- We have calculated the resonant frequency using Akamatsu’s equations for the 7L tank (foam on the walls, water volume = 20cm height x 18cm width x 19cm depth) where we have recorded the 516 simple sounds. We found a frequency of resonance of 6856 Hz for mode (1,1,1) and 9954 Hz for mode (2,1,1). The clocs and sharp clicks have frequencies largely under these resonant frequencies

(mean cloc frequency = 223 Hz in cavefish; mean sharp click frequency = 351 Hz in cavefish) so we decided to not apply a filter on them. As regarding the clicks, the situation is a little bit more complex. In the wild, in large pools where the phenomenon of resonance can be neglected, many of the best-recorded clicks (the closest to the hydrophone) had some components above 13000-14000 Hz (due to the nature of the sound, bubble crush), so higher than the resonant frequencies calculated for the 7L tank. So we decided to not filter the clicks obtained in the 7L aquarium even if a part of the signal may be due to some resonance phenomenon.

In the ACP, it seems authors did not test the correlation between data to avoid overestimation? If features are correlated, they have to be removed.

> Yes, we have produced the correlation matrix of the 9 variables used in the PCA and tested the impact of correlated variables as demanded by the guidelines for statistics in Nature Communications.

Below is the detailed procedure:

We first performed a PCA using the whole set of variables (n=9).

We obtain the scree plot below (scree plot #1).

	eigenvalue	percentage of variance
comp 1	5.68	63.14
comp 2	1.21	13.39
comp 3	0.97	10.78
comp 4	0.60	6.69
comp 5	0.29	3.17
comp 6	0.14	1.54
comp 7	0.08	0.85
comp 8	0.04	0.44
comp 9	0.00	0.00

We then analyzed the correlation matrix for the 9 variables. We noticed that some of them were strongly correlated together, especially max, bandwidth and quartile 50:

	Duration	PeakFreq	Amplitude	Min	Max	Bandwidth	Quart25	Quart50	Quart75
Duration	1.0000000	-0.1314887	0.10900370	-0.08534355	-0.1862395	-0.1859572	-0.1664121	-0.1409741	-0.09910302
PeakFreq	-0.13148865	1.0000000	-0.25820455	0.41441760	0.6309346	0.6286992	0.8857542	0.7231983	0.58302455
Amplitude	0.10900370	-0.2582045	1.0000000	0.02973185	-0.6928383	-0.6952369	-0.4754383	-0.6571852	-0.79958989
Min	-0.08534355	0.4144176	0.02973185	1.0000000	0.3047330	0.2956392	0.4025060	0.3025427	0.32426571
Max	-0.18623955	0.6309346	-0.69283834	0.30473296	1.0000000	0.9999545	0.8308994	0.8695275	0.83256393
Bandwidth	-0.18595723	0.6286992	-0.69523688	0.29563922	0.9999545	1.0000000	0.8293720	0.8691219	0.83182150
Quart25	-0.16641207	0.8857542	-0.47543831	0.40250603	0.8308994	0.8293720	1.0000000	0.9024882	0.76302507
Quart50	-0.14097409	0.7231983	-0.65718518	0.30254271	0.8695275	0.8691219	0.9024882	1.0000000	0.87493400
Quart75	-0.09910302	0.5830246	-0.79958989	0.32426571	0.8325639	0.8318215	0.7630251	0.8749340	1.0000000

We therefore re-did the PCA using 7 variables, discarding max and bandwidth. We obtained a new scree plot (scree plot #2).

	eigenvalue	percentage of variance
comp 1	3,99	56,95
comp 2	1,17	16,77
comp 3	0,97	13,84
comp 4	0,60	8,56
comp 5	0,15	2,10
comp 6	0,08	1,18
comp 7	0,04	0,60

When compared to the first analysis, it appears that the relative contribution of the three first components (comp1 comp2 comp3) to the global variance did not strongly change so we decided to keep the first analysis.

Line 310. « Sounds used for these analyses were high pass filtered at 150 Hz. » So how is it possible to have min freq that are below 150Hz ?

> The high-pass filter used (FIR with Hann window –also referred as “Hanning” window) is not straight, it lets a part of the signal pass under 150 Hz.

Table 1. duration, why 10^{-3} ? it would be 0.00X ms ? Corrected, thanks for noting the error.

Reviewers' Comments:

Reviewer #3:

Remarks to the Author:

Different modifications and clarifications were done.

To answer a question on the similarity between lab- and wild recorded sounds, authors have extracted the physical parameters of sounds and performed a PCA. They use many frequency parameters they can find in a single data (Quartile - Bandwidth -...). I simply do not know if quartiles can be used in PCA. Sorry.

I am still circumspect for some of the sounds because I do not understand the fish interest to produce frequencies that cannot be detected. As an author, I would have published only the data on the sounds (sharp click) that elicit a behaviour and kept the other sounds for additional experiments. Moreover, it would focus on the main message of the paper. Of course, I am not the author.

Authors state: "a given context (here, solo, or duo, or group) encompasses different behaviors, hence it is expected that different sounds can be produced. For example, in a group agonistic behaviors, establishment of hierarchy, schooling, kin/non-kin recognition, sleeping, exploration and locomotion, or else reproductive behaviors, can all arise." Could you please cite a paper showing different kinds of sounds were done during the same behavioural context?

Response to Reviewer 3

Different modifications and clarifications were done.

To answer a question on the similarity between lab- and wild recorded sounds, authors have extracted the physical parameters of sounds and performed a PCA. They use many frequency parameters they can find in a single data (Quartile - Bandwith -...). I simply do not know if quartiles can be used in PCA. Sorry.

> yes we have used the dataset appropriately.

I am still circumspect for some of the sounds because I do not understand the fish interest to produce frequencies that cannot be detected. As an author, I would have published only the data on the sounds (sharp click) that elicit a behaviour and kept the other sounds for additional experiments. Moreover, it would focus on the main message of the paper. Of course, I am not the author.

> the editor and the authors disagree with this suggestion to delete a lot of data from the paper. We have added a note, as suggested by the editor, line 163.

Authors state: "a given context (here, solo, or duo, or group) encompasses different behaviors, hence it is expected that different sounds can be produced. For example, in a group, agonistic behaviors, establishment of hierarchy, schooling, kin/non-kin recognition, sleeping, exploration and locomotion, or else reproductive behaviors, can all arise." Could you please cite a paper showing different kinds of sounds were done during the same behavioural context?

> For example, Nowiki and Marler, 1988, in "How do birds sing?", have shown in birds (equipped with a syrx) that some species can produce different sound types, temporally very close to each other, in a given sequence. This suggests that the animal can produce different sounds in a given behavioral context. And this is even more likely to occur in species which have several sound producing organs or mechanisms, like fish.

The statement quoted by the reviewer was in our response to reviewer's comments, not in the manuscript text. Therefore, to avoid adding this side-discussion to the text, the statement has not been added in the paper and the reference is given as an example but has not been added in the manuscript either.